# Symmetric Reinforcement Learning Loss for Robust Learning on Diverse Tasks and Model Scales

Ju-Seung Byun [1]   Andrew Perrault [1]

## Abstract

Reinforcement learning (RL) training is inherently unstable due to factors such as moving targets and high gradient variance. Reinforcement Learning from Human Feedback (RLHF) and Reinforcement Learning from AI Feedback (RLAIF) introduce additional challenges. For instance, diverse preferences complicate the alignment process, and prediction errors in a trained reward model can become more severe as the LLM generates unseen outputs. These RL challenges create confusion about whether the probability of an action for a given state should be increased or decreased, similar to the noise in labels for classification tasks. In this work, we focus on RL algorithms that share learning difficulties with cross-entropy loss, especially for low-probability predictions. To enhance stability, we adapt reverse cross-entropy (RCE) from supervised learning for noisy data, defining a symmetric RL loss. We demonstrate performance improvements across various tasks and scales. We conduct experiments in discrete action tasks (Atari games) and continuous action space tasks (MuJoCo benchmark and Box2D) using Symmetric A2C (SA2C) and Symmetric PPO (SPPO). Notably, SPPO shows strong performance across different hyperparameters. Furthermore, we validate the symmetric RL loss in the RLHF framework using PPO for natural language processing tasks such as IMDB positive sentiment and TL;DR summarization.

[1]Department of Computer Science and Engineering, The Ohio State University. Correspondence to: Ju-Seung Byun <byun.83@osu.edu>.

*Proceedings of the 42^nd International Conference on Machine Learning*, Vancouver, Canada. PMLR 267, 2025. Copyright 2025 by the author(s).

## 1. Introduction

Recent advancements in Large Language Models (LLMs) have shown impressive performance across various natural language processing tasks (Chung et al., 2022; Wei et al., 2023), robot control (Huang et al., 2022; Driess et al., 2023), and healthcare (Lee et al., 2023c; Huang et al., 2020). However, as these LLMs are typically trained to predict the next word in a provided dataset, they require post-training processing to make them useful for particular tasks. Reinforcement Learning from Human Feedback (RLHF) trains LLMs to generate responses aligned with user preferences through human feedback. Additionally, Reinforcement Learning from AI Feedback (RLAIF), which leverages feedback from well-trained AI models, has also been employed (Lee et al., 2023a; Bai et al., 2022). Thus, adapting fundamental Reinforcement Learning (RL) algorithms such as REINFORCE (Williams, 1992), A2C (Mnih et al., 2016), and PPO (Schulman et al., 2017) to suit the fine-tuning of LLMs for LLM tasks is an area of active interest (Ahmadian et al., 2024; Ouyang et al., 2022; Rafailov et al., 2023).

RL methods (Sutton et al., 2000; Sutton & Barto, 2018a) have lead to substantial breakthroughs in tasks such as robot control and game playing. Still, they entail learning instability compared to supervised learning due to factors such as moving targets, high-gradient variance, and training value functions. The RL literature has proposed various methods to make the RL process more robust, such as preventing overestimation with Double DQN (van Hasselt et al., 2015), reducing variance with Generalized Advantage Estimation (GAE) (Schulman et al., 2018), updates within the trust region (Schulman et al., 2015; 2017), and encouraging diverse behavior with Soft Actor-Critic (SAC) (Haarnoja et al., 2018). In addition to the methods devised specifically for RL problems, RL literature has also adopted supervised learning techniques to make the learning process more robust. For example, ensembles have been used for more accurate value function prediction, while Layer Normalization and Batch Normalization have been employed to constrain predictions for out-of-distribution samples, thereby mitigating the overestimation and extrapolation.

RLHF (Ouyang et al., 2022; Lee et al., 2023b) and RLAIF (Lee et al., 2023a; Bai et al., 2022; Byun et al., 2024) po-

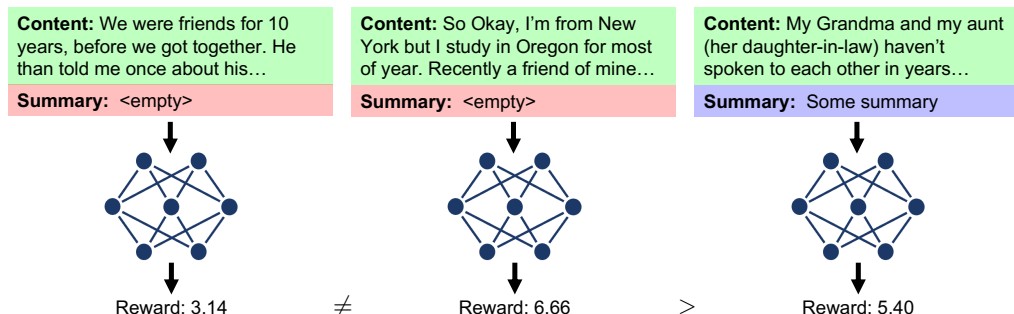

*Figure 1.* Example of reward prediction errors in a trained reward model for TL;DR summarization. The generated summary samples (left and middle) are both *empty*, yet they receive significantly different rewards. The middle sample is higher than some summarization (right) and even scores higher (6.66) than the average reward score of SPPO (6.13). The full text for these samples can be found in Appendix 15.

tentially introduce additional training challenges. For example, these algorithms often receive feedback from multiple sources (humans or AI models) to align LLMs, and each feedback provider may have different preferences, meaning a sample considered preferable by one provider could be deemed undesirable by another (Ethayarajh et al., 2024; Chakraborty et al., 2024). In addition, RLHF and RLAIF often leverage a trained reward model to provide feedback on samples generated by the LLM. This indirection raises the question: *does the learned reward model provide the correct reward?* The reward model has prediction errors itself (See Figure 1), but as the LLM is trained with RL, its outputs deviate from the reward model's training dataset, introducing more errors (noise) in the reward model's predictions for out-of-distribution samples.

The challenges associated with RL, RLHF, and RLAIF, as mentioned above, can introduce confusion when calculating advantage values in RL algorithms like A2C and PPO. Specifically, an action that should have a positive advantage value may have a negative sign in the next update, depending on which samples (states, actions) are generated and how the batch is composed during advantage normalization. The sign of the advantage determines whether the probability of a corresponding action for a given state increases or decreases in policy gradient algorithms. If the advantages are predicted incorrectly, this can lead to learning in the opposite direction. We hypothesize that these difficulties are similar to noisy classification tasks in supervised learning, where some labels are incorrect.

In this paper, we leverage a technique developed for classification tasks with noisy labels, employing a robust loss function to enhance the learning procedures of A2C and PPO. We define a symmetric RL loss, whose fundamental mechanism aligns with the robust loss function used in supervised learning (Wang et al., 2019), to improve the robustness of RL procedure for A2C and PPO (See Section 4.3). We apply this symmetric RL loss to A2C and

PPO, naming them Symmetric A2C (SA2C) and Symmetric PPO (SPPO), and evaluate their performance across various tasks and model scales.

First, we assess the performance gains of SA2C and SPPO on Atari games (Mnih et al., 2016), which have discrete action spaces, as well as on the MuJoCo benchmark (Todorov et al., 2012) and Box2D (Catto, 2011) environments, which have continuous action spaces. For these control tasks, we introduce a noisy reward variant, hypothesizing that it will increase confusion in advantage prediction to better evaluate our method. Additionally, we test our method on RLHF tasks using LLMs, such as IMDB positive sentiment analysis (Maas et al., 2011) and TL;DR summarization (Völske et al., 2017). The IMDB task involves generating positive sentiment for a given context and TL;DR is a summarization task where an LLM is required to summarize content.

SA2C and SPPO demonstrate better performance improvements across diverse control tasks compared to A2C and PPO. Notably, both SA2C and SPPO perform well in settings with added noise to the reward. Additionally, SPPO shows consistent performance improvements across various hyperparameters (Table 11). We analyze why SPPO exhibits more robust improvements than SA2C in Section 5.4. Furthermore, SPPO shows superior performance to PPO in RLHF tasks, such as IMDB positive sentiment and TL;DR summarization. We demonstrate that SPPO outperforms PPO on reward in both tasks, and SPPO's summarization is significantly better, as measured by the win rate judged by GPT-4 Turbo (`gpt-4-turbo-2024-04-09`).

In summary, our key contributions are:

- We propose the symmetric RL loss for A2C and PPO, along with the gradient analysis that aligns with the gradient behavior of robust loss functions used in noisy classification tasks in Section 4.3.

- We conduct experiments across various environments

and model scales, demonstrating performance improvements to validate the symmetric RL loss for general control tasks and RLHF tasks in Section 5.

- We analyze how PPO can introduce additional confusion in advantage estimates, which justifies using symmetric RL loss (See Section 5.4). This shows that SPPO demonstrates consistent improvement across a range of hyperparameters.

## 2. Related Work

We introduce robust loss functions studied in the context of noise in supervised learning classification tasks. Ghosh et al. (2017) prove that, in the presence of a noisy dataset, the mean absolute error (MAE) has a slower learning speed compared to cross-entropy loss (CE), but the model learns more robustly. Zhang & Sabuncu (2018) propose a generalized cross entropy loss $L_q$, which becomes CE when $q \to 0$, and becomes MAE when $q \to 1$. By adjusting this parameter $0 \leq q \leq 1$, robust learning is achieved in noisy datasets. The symmetric cross entropy (SCE) (Wang et al., 2019) that we mainly refer to suggests a symmetric cross-entropy loss. This loss not only considers the flow of information from the true distribution to the model's predictions but also incorporates information flowing in the reverse direction. SCE works better than GCE in general, especially for data with high noise rates. Ma et al. (2020) introduce various loss functions and classify them into types: *Active Loss* and *Passive Loss* functions. They demonstrate that normalizing the loss can help improve robustness. They use a combination of one active loss and one passive loss like SCE. We define a loss function that considers reverse information to match the RL version.

In the RL literature, Wang et al. (2018) propose using a confusion matrix to handle perturbed rewards, predicting surrogate rewards for robust policy updates. While this method appears effective for Atari games, later research (Chen et al., 2024) shows that it does not outperform corresponding baselines in continuous tasks. Additionally, introducing noise in RL has demonstrated performance benefits. For instance, Obando-Ceron et al. (2023) show that smaller batch sizes improve performance, and Schaul et al. (2022) present that policy churn aids exploration. These studies primarily conduct experiments on Atari games, which require navigating many novel states. However, whether noise is beneficial or not in continuous action spaces remains debatable (Mai et al., 2022; Byun & Perrault, 2024). Our work proposes a robust loss function designed to handle noise (confusion in advantage prediction) without judging whether the noise is beneficial or not.

Reinforcement Learning from Human Feedback (RLHF) Ouyang et al. (2022); Lee et al. (2023b) and Reinforcement Learning from AI Feedback (RLAIF) (Lee et al., 2023a; Bai et al., 2022) have contributed to the success of large language models (LLMs) by aligning them with user preferences. However, these methods require training a reward model and a value function. Each of these components has prediction errors, and finding appropriate hyperparameters for training requires significant effort. Direct Preference Optimization (DPO) (Rafailov et al., 2023) eliminates the cost associated with the reward model by rearranging PPO loss for ranking-based feedback (e.g., sample A is preferred over sample B). Ethayarajh et al. (2024) remove the requirement ranking-based feedback by modifying DPO loss further, allowing a model to be trained with bad or good labels. Additionally, Chakraborty et al. (2024) demonstrate that feedback from diverse people, each with different preferences, makes a single reward model difficult to reflect preferences correctly. Recent studies focus on sentence-level feedback (Lightman et al., 2023; Wang et al., 2024), but DPO and KTO cannot utilize sentence-level feedback. Therefore, we propose the reverse RL loss term, which can make PPO in existing RLHF methods more robust.

## 3. Preliminaries

### 3.1. Reinforcement Learning

Reinforcement Learning (RL) formulates a Markov decision process (MDP) (Puterman, 2014; Sutton & Barto, 2018b) defined by the tuple $\mathcal{M} = (\mathcal{S}, \mathcal{A}, \mathcal{P}, R, \gamma, \mu)$. At each timestep $t$, an action $a_t \in \mathcal{A}$ is sampled from an agent's policy $\pi_\theta(\cdot \mid s_t)$ for a given state $s_t \in \mathcal{S}$. For the taken action $a_t$, the reward function returns a reward $\mathcal{R}(s_t, a_t)$ where $\mathcal{R} : \mathcal{S} \times \mathcal{A} \to \mathbb{R}$, and the transition probability $\mathcal{P}(\cdot \mid s_t, a_t)$ determines the next state $s_{t+1}$. $\gamma$ is the discount factor, and $\mu$ represents the initial state distribution for $s_0$. The RL objective is to find the optimal $\theta$ that maximizes the expected discounted sum of rewards:

$$\theta^* = \underset{\theta}{\arg\max} \underset{\substack{s_0 \sim \mu \\ a_t \sim \pi_\theta(\cdot \mid s_t) \\ s_{t+1} \sim \mathcal{P}(\cdot \mid s_t, a_t)}}{\mathbb{E}} \left[ \sum_{t=0}^{\infty} \gamma^t R(s_t, a_t) \right]. \quad (1)$$

### 3.2. A2C and PPO Algorithms

The Advantage Actor-Critic (A2C) algorithm (Mnih et al., 2016) is an actor-critic method that combines value-based and policy-based approaches. A2C uses the advantage function $A$ to reduce the variance in policy updates. The policy $\pi_\theta$ is updated by following the gradient of the objective function to maximize the sum of rewards (Equation 1):

$$\nabla_\theta J(\pi_\theta) = \sum_{t=0} \nabla_\theta \log \pi_\theta(a_t \mid s_t) A(s_t, a_t) \quad (2)$$

Proximal Policy Optimization (PPO) (Schulman et al., 2017) aims to update the policy within a trust region. This is

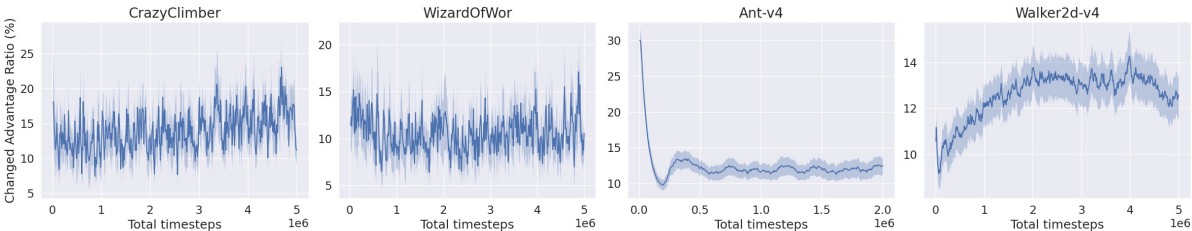

*Figure 2.* Change of advantage rate (%): The graphs show how often the advantage signs flip in various environments as training progresses. In Atari games, often over 5% of samples change signs, while in MuJoCo tasks, usually over 10% of samples change signs after the advantage normalization. We use 5 different random seeds for CrazyClimber and WizardOfWor, and 30 different random seeds for Ant-v4 and Walker2d-v4. The line is the mean of the change ratio across the seeds, and the shaded area represents standard errors.

achieved through a clipped loss function to ensure that the new policy does not deviate too much from the old policy. The PPO loss function can be written as:

$$L_{\text{ppo}}(\theta) = \mathbb{E}_t \left[ \min \left( r_t(\theta) A_t, \text{clip}(r_t(\theta), 1 - \epsilon, 1 + \epsilon) A_t \right) \right] \tag{3}$$

where $r_t(\theta) = \frac{\pi_\theta(a_t|s_t)}{\pi_{\theta_{\text{old}}}(a_t|s_t)}$ is the probability ratio, and $\epsilon$ is a small hyperparameter that controls the range of the clipping. The advantage function estimates how much better an action $a$ is compared to the other actions for at a given state $s$. Both algorithms increase the probability of $a$ for $s$ if the corresponding advantage $A(s,a) > 0$ and decrease it if $A(s,a) < 0$. In the next section, we introduce the connection between A2C and PPO with the cross-entropy loss for classification and define the symmetric RL loss.

### 3.3. Symmetric Cross Entropy

Symmetric Cross Entropy (SCE) (Wang et al., 2019) is designed for noisy classification datasets. Cross Entropy (CE) loss (Equation 4) performs effectively when the data is clean; however, it encounters challenges in the presence of noise. Given a true distribution $q$ and a predicted distribution $p$, $p$ is learned based on the information derived from $q$ according to information theory. However, when $q$ is noisy, $p$ can only approximate the true distribution to a limited extent. To address this issue, SCE incorporates information in the opposite direction through Reverse Cross Entropy (RCE) (Equation 5).

$$L_{\text{ce}} = -\sum_{k=1}^{K} q(k|\mathbf{x}) \log p(k|\mathbf{x}) \tag{4}$$

$$L_{\text{rce}} = -\sum_{k=1}^{K} p(k|\mathbf{x}) \log q(k|\mathbf{x}) \tag{5}$$

where $k \in \{1, \ldots, K\}$ is a class and $\mathbf{x}$ is an input. RCE loss has been proven to be robust to a certain amount of noise, but the learning speed is too slow. Therefore, SCE

combines CE and RCE losses (Equation 6),

$$L_{\text{sce}} = \alpha L_{\text{ce}} + \beta L_{\text{rce}} \tag{6}$$

where $\alpha$ and $\beta$ are constants determining the contribution of each part. SCE demonstrates performance improvement across various noisy ratios and types. As mentioned in the introduction section, the RL training process can lead to noisy advantage predictions, so we propose a symmetric RL loss in the next approach section.

## 4. Approach

This section introduces the reverse RL loss and proposes the symmetric RL loss for A2C (Mnih et al., 2016) and PPO (Schulman et al., 2017), an RL version of Symmetric Cross Entropy (SCE) (Wang et al., 2019). A2C and PPO training procedures basically increase or decrease the probability of an action depending on the advantage sign, but the advantage prediction involves noise due to several factors. A highly engineered reward function is required to eliminate errors, and the trained reward model has prediction errors in RLHF (Ouyang et al., 2022) and RLAIF (Lee et al., 2023a; Bai et al., 2022). Receiving feedback from multiple sources further complicates the training of the reward model (Chakraborty et al., 2024). Additionally, the value function also has estimation errors, and the sign of the advantage in advantage normalization depends on how the batch is composed. PPO increases sample efficiency compared to A2C, but the off-policy part can introduce confusion in advantage predictions (See Section 5.4). Similar to SCE, which is robust to noisy data, the symmetric RL loss enhances robustness in an RL environment that can introduce noise.

### 4.1. Reverse Reinforcement Learning Loss

Given a true (target) distribution $q$ and a predicted distribution $p$, if $q$ is noisy, training $p$ can be challenging and $p$ cannot accurately reflect the true distribution. Reverse Cross Entropy (RCE) considers the reverse information from $p$.

We propose that the reverse RL losses for A2C and PPO also incorporate reverse information to address noisy factors in the RL training procedure. The RCE loss (Equation 5) defines $\log 0 = Z$ where $Z < 0$ is some constant for $q(k|\mathbf{x}) = 0$ (Wang et al., 2019). We also use this definition for the negative advantage and this is also useful to prove the robustness of the reverse RL losses. For all tasks conducted in this paper, we use $Z = -1$. Note that the constant terms $Z$ and $\beta$ in Equation 7 and 9 are multiplied together, so we control the impact of the reverse RL loss solely by adjusting $\beta$. For example, $(\beta = 1.0, Z = -1.0)$ and $(\beta = 10.0, Z = -0.1)$ yield the exact same results.

Suppose there exist $k$ actions and $a^{(i)}$ indicates $i^{\text{th}}$ action. $\pi_\theta^{(i)} = \pi_\theta(a^{(i)}|s)$ for a state $s$. Let's denote the possible action probabilities set $s$ as $\pi_\theta(s) = \{\pi_\theta^{(1)}, \pi_\theta^{(2)}, ..., \pi_\theta^{(k)}\}$. Note that we discretize the continuous action space for continuous action tasks (Tang & Agrawal, 2020). One thing we need to note is that when updating a policy, we use advantages instead of label sets in RL. *Advantages can have negative values* (negative labels) unlike ordinary labels. We only consider the sign of the advantage[1] because this advantage is the role of the label in supervised learning. For a sampled action probability $\pi_\theta^{(i)}$ and the corresponding advantage $A(s, a^{(i)}) = A^{(i)}$, the *sample-wise reverse A2C (RA2C) loss* is:

$$L_{\text{ra2c}}(\pi_\theta(s), A^{(i)}) = \begin{cases} \sum_{j \in [k] \setminus \{i\}} -\pi_\theta^{(j)} A^{(i)} Z, \\ \quad \text{if } A^{(i)} > 0 \\ \sum_{j \in [k] \setminus \{i\}} \pi_\theta^{(j)} A^{(i)} Z, \\ \quad \text{if } A^{(i)} < 0 \end{cases} \quad (7)$$

For a positive advantage $A$, the difference between A2C's loss $A \log \pi$ and CE loss $1 \log p$ is that A2C can be considered as CE multiplied by the advantage. In terms of gradients, $A$ is a constant, so A2C reflects the information $A$ times more strongly than the CE loss. Thus, we also reflect the reverse direction $A$ times more strongly. Similarly, since PPO has $\pi_{\text{old}}^{(i)}$ term in the loss, the sample-wise reverse PPO (RPPO) loss just introduces the additional constant $\pi_{\text{old}}^{(i)}$ for a sampled action probability $\pi_\theta^{(i)}$ to consider the same amount of reverse information:

$$L_{\text{rppo}}(\pi_\theta(s), A^{(i)}, \pi_{\text{old}}^{(i)}) = \begin{cases} \sum_{j \in [k] \setminus \{i\}} -\frac{\pi_\theta^{(j)} A^{(i)} Z}{\pi_{\text{old}}^{(i)}}, \\ \quad \text{if } A^{(i)} > 0 \\ \sum_{j \in [k] \setminus \{i\}} \frac{\pi_\theta^{(j)} A^{(i)} Z}{\pi_{\text{old}}^{(i)}}, \\ \quad \text{if } A^{(i)} < 0 \end{cases} \quad (8)$$

We define the symmetric RL loss, which consists of the original RL loss (A2C or PPO) and the corresponding reverse

---

[1] We do not consider cases where the advantages are zero because they do not affect policy updates.

RL loss, in Section 4.2. We then analyze how these reverse RL losses contribute to RL robustness in Section 4.3.

## 4.2. Symmetric Reinforcement Learning Loss

The *Symmetric Reinforcement Learning (SRL) loss* $L_{\text{srl}}$ consists of two parts like SCE (Equation 6): the original actor loss $L_{\text{rl}}$ (A2C or PPO) and the corresponding reverse RL loss $L_{\text{rev}}$ (RA2C or RPPO). $L_{\text{srl}}$ flexibly adjusts the symmetric learning framework with two additional hyperparameters ($\alpha > 0$ and $\beta > 0$) as follows:

$$L_{\text{srl}} = \alpha L_{\text{rl}} + \beta L_{\text{rev}} \quad (9)$$

We name A2C and PPO using the symmetric RL loss as *Symmetric A2C (SA2C)* and *Symmetric PPO (SPPO)*, respectively. The meanings of $\alpha$ and $\beta$ align with SCE, where $\alpha$ represents the degree of actively training a policy, and $\beta$ serves as auxiliary support to stabilize the entire learning process. In the following section, we analyze the gradient of the two types of losses.

## 4.3. Gradient Analysis

For an input $\mathbf{x}$ and the corresponding correct label $k$, the cross entropy (CE) loss gradient is $-\frac{1}{p_\theta(k|\mathbf{x})} \nabla_\theta p_\theta(k|\mathbf{x})$. Smaller $p_\theta$ values aggressively increase the magnitude of the gradient. CE loss rapidly increases uncertain predictions. If there is no noise, this method is correct, but it may lead to incorrect predictions on noisy datasets and excessive overfitting (Zhang & Sabuncu, 2018). A2C and PPO losses also have the same issue. For A2C, the gradient is simply multiplied by an advantage $A$, i.e., $-\frac{A(s,a)}{\pi_\theta(a|s)} \nabla_\theta \pi_\theta(a|s)$. In the case of PPO, the magnitude of the gradient tends to increase as the probability of an action decreases. Consider a sample that passes the clipping function: the difference between $\pi_{\text{old}}$ and $\pi$ is within the $\epsilon$ bound. As the denominator $\pi_{\text{old}}$ gets smaller, the magnitude of the gradient increases.

**Detailed Analysis:** The symmetric RL loss gradient analysis aligns with the analysis of SCE. For simplicity, we set $\alpha$ and $\beta$ to 1 and examine the gradient direction for two types of A2C loss (RL and reverse RL) with respect to the action logits $z$. We use the notation defined in Section 4.1 and introduce the case when $A^{(i)} > 0$. For the full derivation including SPPO and $A^{(i)} < 0$, please refer to Appendix A. The sample-wise SA2C loss is:

$$L_{\text{sa2c}} = L_{\text{a2c}} + L_{\text{ra2c}} \quad (10)$$

The gradients for each part are:

$$\frac{\partial L_{\text{a2c}}(\pi^{(i)}, A^{(i)})}{\partial z_y} = \begin{cases} A^{(i)}(\pi^{(i)} - 1), & \text{if } i = y \\ A^{(i)} \pi^{(y)}, & \text{if } i \neq y \end{cases} \quad (11)$$

*Table 1.* Mean final scores and standard errors (over the last 10 episodes) of PPO and SPPO on Atari games, without and with binary symmetric channel (BSC) noise with a crossover probability of 0.1 across 5 seeds. Full results can be found in Table 10.

| | **WITHOUT NOISE** | | $\epsilon \sim \mathbf{BSC}(0.1)$ | |
| | PPO | **SPPO** | PPO | **SPPO** |
|---|---|---|---|---|
| ALIEN | **1128 ± 105** | 1081 ± 79 | 525 ± 26 | **713 ± 26** |
| CENTIPEDE | 2961 ± 379 | **3694 ± 224** | 4759 ± 257 | **7525 ± 769** |
| CRAZYCLIMBER | 86764 ± 3568 | **103588 ± 2871** | 71144 ± 11060 | **99810 ± 2487** |
| GRAVITAR | 371 ± 47 | **442 ± 67** | 269 ± 39 | **332 ± 61** |
| QBERT | 4352 ± 128 | **4412 ± 282** | 2827 ± 1927 | **4020 ± 2415** |
| MSPACMAN | 837 ± 62 | **1204 ± 86** | 704 ± 41 | **1011 ± 52** |

$$\frac{\partial L_{\mathrm{ra2c}}(\pi^{(i)}, A^{(i)})}{\partial z_y} = \begin{cases} -A^{(i)}Z\pi^{(y)}(\pi^{(y)} - 1), \\ \quad \text{if } i = y \text{ and } A^{(i)} > 0 \\ -A^{(i)}Z\pi^{(y)}\pi^{(i)}, \\ \quad \text{if } i \neq y \text{ and } A^{(i)} > 0 \end{cases} \quad (12)$$

Thus, the SA2C loss gradient is:

$$\frac{\partial L_{\mathrm{sa2c}}}{\partial z_y} = \begin{cases} \underbrace{A^{(i)}(\pi^{(i)} - 1)}_{\nabla L_{\mathrm{a2c}} < 0} \underbrace{-A^{(i)}Z\pi^{(i)}(\pi^{(i)} - 1)}_{\nabla L_{\mathrm{ra2c}} < 0}, \\ \quad \text{if } i = y \text{ and } A^{(i)} > 0 \\ \underbrace{A^{(i)}\pi^{(y)}}_{\nabla L_{\mathrm{a2c}} > 0} \underbrace{-A^{(i)}Z\pi^{(y)}\pi^{(i)}}_{\nabla L_{\mathrm{ra2c}} > 0}, \\ \quad \text{if } i \neq y \text{ and } A^{(i)} > 0 \end{cases}$$

$$(13)$$

For both cases, the gradient directions of the RL (A2C) loss and the reverse RL (RA2C) loss are aligned. When $i = y$ and $A^{(i)} > 0$, the gradient of the RA2C loss is $-A^{(i)}Z\pi^{(y)}(\pi^{(y)} - 1)$, reaching its maximum magnitude at $\pi^{(y)} = 0.5$ as a parabolic function. This means that the accelerator helps the probability $\pi^{(i)}$ increase most rapidly when the action to take is ambiguous. When $i \neq y$ and $A^{(i)} > 0$, the probability of actions other than $a^{(i)}$ is reduced, and this reduction is influenced by the confidence of both $\pi^{(i)}$ and $\pi^{(y)}$. Specifically, the gradient of the RA2C loss is $-A^{(i)}Z\pi^{(y)}\pi^{(i)}$. When both $\pi^{(i)}$ and $\pi^{(y)}$ are 0.5, representing the most ambiguous predictions, the accelerator aids the A2C loss in reducing $\pi^{(y)}$ most effectively. Thus, the RA2C loss helps deviate from ambiguous predictions as an accelerator. SPPO's loss gradients are also aligned like SA2C and follow the same mechanism (See Appendix B.2).

# 5. Experiments

To validate the effectiveness of our algorithm, we conduct experiments on various tasks and models of different scales. First, we experiment on Atari games (Mnih et al., 2013) featuring discrete action spaces (Section 5.1), as well as MuJoCo benchmark tasks (Todorov et al., 2012) and Box2D

tasks (Catto, 2011) (Section 5.2) with continuous action spaces using Stable-Baselines3 (Raffin et al., 2021). In these control tasks, we also create a variant of each that introduces reward noise, hypothesizing that it will create more confusion in advantage prediction. SPPO performs better than SA2C for various reverse RL loss hyperparameters $\beta$. We also evaluate our method on IMDB and TL;DR datasets using TRIL (Chang et al., 2023) to determine whether our approach is practical for LLM tasks. We primarily present the experimental results for PPO in the main paper. In the latter part of this section, we analyze why our method works better with PPO than A2C (Section 5.4), conduct hyperparameter sensitivity tests, and examine the training cost (Section 5.5).

## 5.1. Discrete Action Space Tasks

We first conduct experiments on Atari games (Mnih et al., 2016) that the action spaces are discrete to evaluate SPPO and SA2C. We primarily select 22 games based on the reported score for A2C in Schulman et al. (2017), focusing on games where the A2C scores are not close to 0, as this allows us to demonstrate meaningful score changes.

To introduce some reward noise, we simply flip the reward from 0 to 1 or from 1 to 0 with a probability of 10%. We denote this noise setting as a Binary Symmetric Channel (BSC). This setting is analogous to a potential problem in ranking-based feedback (Ouyang et al., 2022) from humans or AI, where evaluators may have different preferences, resulting in reversed scores. We observe that SA2C shows marginal improvements (Table 7), with a narrow range of effective hyperparameter $\beta$ values. In contrast, SPPO performs well in both noise-free and noisy environments (See Section 5.4 for discussion). Table 1 presents partial results, while the complete results for SPPO, including training curves (Figure 4), can be found in Table 10. SPPO achieves 16 out of 22 wins in noise-free settings and 19 out of 22 wins in noisy settings.

*Table 2.* Mean final scores and standard errors (over the last 10 episodes) of PPO and SPPO on Atari games, without and with binary symmetric channel (BSC) noise with a crossover probability of 0.1 across 5 seeds. To leverage the reverse RL loss, we discretize the continuous action space. DPPO is added as another baseline ($\alpha = 1.0$, $\beta = 0.0$), and DSPPO is our proposed method. Full results can be found in Table 12 and 13.

| $\epsilon \sim \mathcal{N}(0, 0.05^2)$ | ANT | HOPPER | HALFCHEETAH | HUMANOIDSTANDUP |
|---|---|---|---|---|
| PPO | $601 \pm 47$ | $1936 \pm 147$ | $2068 \pm 208$ | $80945 \pm 2130$ |
| DPPO | $1897 \pm 86$ | $2153 \pm 106$ | $2722 \pm 188$ | $\mathbf{146038 \pm 1841}$ |
| **DSPPO** | $\mathbf{2095 \pm 102}$ | $\mathbf{2333 \pm 109}$ | $\mathbf{3118 \pm 195}$ | $145974 \pm 2520$ |
| | WALKER2D | SWIMMER | BIPEDALWALKER | LUNARLANDERCONTINUOUS |
| PPO | $1270 \pm 107$ | $44 \pm 3.0$ | $158 \pm 15.2$ | $181 \pm 13.8$ |
| DPPO | $3419 \pm 100$ | $57 \pm 3.6$ | $\mathbf{274 \pm 7.1}$ | $281 \pm 5.7$ |
| **DSPPO** | $\mathbf{3523 \pm 129}$ | $\mathbf{72 \pm 5.1}$ | $267 \pm 8.8$ | $\mathbf{294 \pm 3.3}$ |

### 5.2. Continuous Action Space Tasks

Next, we perform experiments on MuJoCo benchmark (Todorov et al., 2012) and Box2D (Catto, 2011) continuous action space environments. To utilize the reverse RL loss, we need other action probabilities for a sampled action probability. However, conventional RL uses a multivariate Gaussian distribution as a policy, so it cannot provide the other action probabilities. Thus, we discretize the continuous action space (Tang & Agrawal, 2020), naming these methods DA2C and DPPO, and add them as additional baseline comparisons.

Note that discretizing the continuous action space generally works better than the original RL methods like A2C and PPO for these tasks if the continuous action space is discretized with a sufficient number of bins. This discretized distribution can represent more complex distributions than a diagonal Gaussian distribution (where the covariance is diagonal). We apply the reverse RL loss to both DA2C and DSPPO.

Since the reward functions in these environments are highly engineered, we perturb the reward function with Gaussian noise with a mean of 0 and a standard deviation of 0.05. Table 2 shows partial results for SPPO under noise settings. The full experiment results are in Table 12 and 13. Similar to the Atari game results, SA2C without noise shows tied performance in the noiseless setting, and improvements when the reward noise is introduced. SPPO consistently shows robust performance gains across a wide range of $\beta$ values for both settings.

### 5.3. RLHF Tasks

The final tasks are RLHF tasks to assess our method's applicability to large language models. The first task, IMDB positive sentiment, aims to generate positive sentiment continuations for movie reviews (Maas et al., 2011). The senti-

ment classifier (Sanh et al., 2019) is used as a reward model to evaluate how positive a provided text is. The base policy is GPT-2 (Radford et al., 2019), which we fine-tune using PPO or SPPO. We evaluate this model based on the reward score and perplexity. SPPO shows improvement in both reward score and perplexity compared to PPO.

The second RLHF task is TL;DR summarization (Völske et al., 2017). The objective is to summarize Reddit posts. The reward model is a fine-tuned GPT-J (Wang & Komatsuzaki, 2021) with LoRA adapters (Hu et al., 2021) by Chang et al. (2023). The training dataset for this reward model is the filtered dataset with additional human preference data used in Stiennon et al. (2020).

The base policy model is an open-source GPT-J model (`CarperAI/openai_summarize_tldr_sft`) with added LoRA adapters. Note that the open-source GPT-J mode often outputs empty summarizations for most evaluation data. Therefore, we report results after 10 epochs of RL updates as an alternative to SFT, as it begins to consistently summarize posts. We evaluate SPPO based on reward score, perplexity, and win rate. This win rate is judged by GPT-4 Turbo (OpenAI, 2024) (`gpt-4-turbo-2024-04-09`) by comparing the generated output and reference text. Even though the perplexity of SPPO is slightly higher than that of PPO, there is an improvement in the reward score and a significantly increased win rate.

We also conduct experiments using Qwen2-0.5B as the policy model with a LoRA adapter, employing the same reward model and hyperparameters: $\alpha = 0.5$ and $\beta = \{0.2, 20.0\}$ for SPPO. Specifically, $\beta = 0.2$ is used for GPT-J, and $\beta = 20.0$ is used for SPPO in the continuous tasks. Overall, SPPO outperforms PPO (Figure 3). Notably, SPPO demonstrates a significant performance boost with $\beta = 20.0$, although its performance drops sharply after 300 epochs. In contrast, PPO and SPPO with $\beta = 0.2$ show a similar performance drop around epoch 900.

*Table 3.* RM Score indicates the reward model score, Perplexity measures the uncertainty of the model, and Win Rate is judged by GPT-4 Turbo by comparing the generated output and reference text. We use 4 different random seeds for each task.

| | IMDB SENTIMENT | | TL;DR SUMMARIZATION | | |
|---|---|---|---|---|---|
| | RM SCORE (↑) | PERPLEXITY (↓) | RM SCORE (↑) | PERPLEXITY (↓) | WIN RATE (↑) |
| SFT | $0.54 \pm 0.00$ | $33.02 \pm 0.09$ | $5.83 \pm 0.02$ | $18.35 \pm 0.02$ | $42.00 \pm 2.58$ |
| PPO | $0.89 \pm 0.02$ | $41.09 \pm 0.43$ | $5.94 \pm 0.08$ | $19.08 \pm 0.17$ | $43.25 \pm 3.82$ |
| **SPPO** | $\mathbf{0.92 \pm 0.01}$ | $40.60 \pm 0.44$ | $\mathbf{6.13 \pm 0.02}$ | $19.27 \pm 0.21$ | $\mathbf{52.50 \pm 2.40}$ |

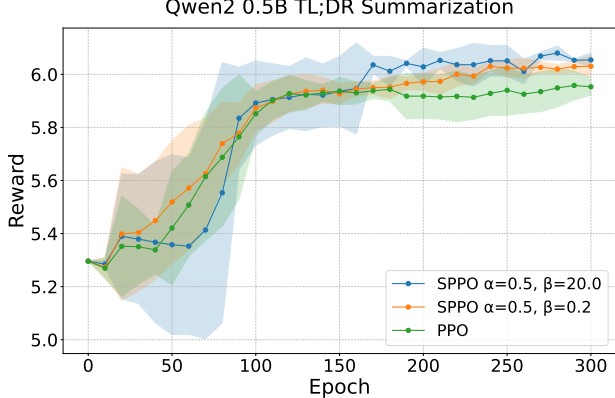

*Figure 3.* TL;DR summarization results with Qwen2-0.5B across 4 different random seeds. SPPO with $\beta = 20.0$ shows a sharp performance surge, followed by a drop after 300 epochs. A similar drop occurs for SPPO with $\beta = 0.2$ and PPO around epoch 900.

In the introduction section, we mention that RLHF or RLAIF have additional errors due to a trained reward model. We check whether the trained reward model in TL;DR has reward prediction errors. Figure 1 shows a dramatic example: the generated summary sample (left) and the middle sample were both *empty*, but their rewards show a huge gap. The middle sample scores (6.66) better than those learned with an SPPO score (6.13). Wrong summaries, like *empty*, can score higher than a summarized text (right). These cases are observed very often. This makes the RL training procedure more noisy and means that the sign of advantage changes depending on how the batch is composed. The more detailed texts for these samples are available in Appendix D.

### 5.4. Why SPPO Works Better Than SA2C

The motivation for using the reverse RL loss is to address the issue of ambiguity in advantage predictions (Section 4.3). We hypothesize that the PPO advantage prediction (sign) is less consistent than in A2C during policy updates, but this does not mean that PPO is worse than A2C. There are two main reasons why consistency is not maintained. First, PPO has improved sample efficiency compared to A2C, but after the first epoch, subsequent updates become

off-policy, affecting advantage estimates. Second, PPO often uses advantage normalization to restrict large advantage values from being involved with policy updates to stabilize the learning process. In addition, PPO often uses smaller mini-batch sizes (e.g., 64), whereas A2C uses the entire dataset for policy updates. Many popular RL code baselines, such as Stable Baselines3 (Raffin et al., 2021), RL4LMs (Ramamurthy et al., 2023), TRL (von Werra et al., 2020), and TRLX (Havrilla et al., 2023) use PPO advantage normalization by default, whereas A2C does not. Our experiments on the usefulness of advantage normalization also show that the performance increase in IMDB is greater than the performance decrease in TL;DR (Appendix 14).

We examine the ratio of advantage sign changes before and after normalization for PPO in Atari games and MuJoCo tasks (Figure 2). This ratio varies across different environments. The advantage sign changes usually exceed 5% for Atari games and 10% for MuJoCo and Box2D environments. These changes introduce the confusion, which makes the reverse RL loss more effective for PPO. This observation aligns with our motivation for using symmetric RL loss to handle noisy data, similar to how it is addressed in noisy classification tasks in supervised learning.

Additionally, since A2C uses the entire dataset (rather than using advantage normalization with small batches) for the policy updates, it introduces less confusion in advantage prediction. As a result, SA2C demonstrates performance comparable to A2C in settings without reward noise (Table 7 and 8), and improvements in settings with reward noise (Table 7 and 9), where advantage estimation is more likely to be confused.

### 5.5. Hyperparameters and Training Cost

Although the symmetric RL loss introduces three additional hyperparameters (Equation 4.2): $\alpha$, $\beta$, and $Z$, we simply fix $\alpha = 0.5$ in all experiments to reduce the overall magnitude of the symmetric loss. Additionally, since $\beta$ and $Z$ are constants that are multiplied together, we can fix one and adjust the other. For example, $(\beta = 1.0, Z = -1.0)$ and $(\beta = 10.0, Z = -0.1)$ yield the same results. In our experiments, we fix $Z = -1$ and adjust $\beta$ to determine the influence of the reverse RL loss.

We test the sensitivity of $\beta$ for SPPO on Atari games with and without noise in the rewards. Table 11 presents the percentage improvements compared to PPO. We exclude excessively large improvements (e.g., 2000%) to avoid skewing the average. These significant improvements typically result from PPO's training failure, while SPPO remains stable (Gopher and WizardOfWor in Figure 4). Fixing $\alpha = 0.5$ and $Z = -1$, we vary $\beta$ and observe consistent improvements, demonstrating SPPO's robustness across hyperparameters. Also, we use the default values of Stable Baselines3 (Raffin et al., 2021) for the other RL hyperparameters; more details can be found in Appendix C.1.

The symmetric RL loss introduces the reverse RL loss term, which is essentially another form of cross-entropy that does not significantly increase training time. In practice, there is no increase in training time for the continuous tasks discussed in Section 5.2 and the LLM tasks in Section 5.3, and a 10–20% increase for the Atari games in Section 5.1.

## 6. Conclusion

We present Symmetric RL loss, inspired by Symmetric Cross Entropy (SCE) (Wang et al., 2019) from supervised learning, to enhance RL robustness. By incorporating reverse information through SCE, we develop SA2C and SPPO, extending standard A2C and PPO algorithms. We test SA2C and SPPO on various discrete and continuous action space tasks and further evaluate SPPO on RLHF tasks like IMDB positive sentiment and TL;DR summarization. Our results show that SPPO consistently outperforms PPO.We attribute this to PPO's off-policy components and advantage normalization with small batch sizes, which cause advantage sign changes (confusion). SCE helps stabilize training, addressing these challenges.

## Acknowledgments

The authors would like to thank the Ohio Supercomputer Center (Center, 1987) for providing the computational resources used in this research.

## Impact Statement

This paper presents work whose goal is to advance the field of Machine Learning. There are many potential societal consequences of our work, none which we feel must be specifically highlighted here.

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

# A. Gradient of RL loss and reverse RL loss

Suppose there exist $k$ actions, and $a^{(i)}$ indicates the $i^{\text{th}}$ action. Let $\pi_\theta^{(i)} = \pi_\theta(a^{(i)}|s)$ denote the policy for a state $s$. The set $\pi_\theta(s) = \{\pi_\theta^{(1)}, \pi_\theta^{(2)}, \ldots, \pi_\theta^{(k)}\}$ represents the possible action probabilities set for $s$. $A^{(i)}$ indicates the corresponding advantage of the sampled action $a^{(i)}$ for $s$. $Z < 0$ is a constant used in the reverse RL loss to handle the computational issue where $\log 0 = -\infty$. For simplicity of notation, we drop $\theta$, $s$, and $a$ from the policy $\pi$. Note that $A^{(i)}$ and $Z$ are not involved with the gradient as they are constants with respect to $\theta$.

## A.1. A2C Loss

The derivation of the A2C loss $L_{\text{a2c}}$ with respect to logits $z$ is presented as follows: For $i = y$,

$$
\begin{aligned}
\frac{\partial \pi^{(i)}}{\partial z_y} &= \frac{\partial}{\partial z_y} \frac{e^{z_i}}{\sum_{w=1}^{k} e^{z_w}} \\
&= \frac{e^{z_i} \sum_{w=1}^{k} e^{z_w} - e^{z_i} e^{z_i}}{(\sum_{w=1}^{k} e^{z_w})^2} \\
&= \pi^{(i)}(1 - \pi^{(i)})
\end{aligned}
\tag{14}
$$

For $i \neq y$,

$$
\begin{aligned}
\frac{\partial \pi^{(i)}}{\partial z_y} &= \frac{\partial}{\partial z_y} \frac{e^{z_i}}{\sum_{w=1}^{k} e^{z_w}} \\
&= -\frac{e^{z_i} e^{z_y}}{(\sum_{w=1}^{k} e^{z_w})^2} \\
&= -\pi^{(i)} \pi^{(y)}
\end{aligned}
\tag{15}
$$

The sample-wise A2C loss is:

$$
L_{\text{a2c}}(\pi^{(i)}, A^{(i)}) = -A^{(i)} \log \pi^{(i)}
\tag{16}
$$

For $i = y$,

$$
\begin{aligned}
\frac{\partial L_{\text{a2c}}}{\partial z_y} &= \frac{\partial}{\partial z_y} - A^{(i)} \log \pi^{(i)} \\
&= -A^{(i)} \frac{\partial}{\partial z_y} \log \pi^{(i)} \\
&= -\frac{A^{(i)}}{\pi^{(i)}} \frac{\partial \pi^{(i)}}{\partial z_y} \\
&= A^{(i)}(\pi^{(i)} - 1) \quad \text{by (14)}
\end{aligned}
\tag{17}
$$

For $i \neq y$,

$$
\begin{aligned}
\frac{\partial L_{\text{a2c}}}{\partial z_y} &= \frac{\partial}{\partial z_y} - A^{(i)} \log \pi^{(i)} \\
&= -A^{(i)} \frac{\partial}{\partial z_y} \log \pi^{(i)} \\
&= -\frac{A^{(i)}}{\pi^{(i)}} \frac{\partial \pi^{(i)}}{\partial z_y} \\
&= A^{(i)} \pi^{(y)} \quad \text{by (15)}
\end{aligned}
\tag{18}
$$

In summary, we have the following form for $L_{\text{a2c}}(\pi^{(i)}, A^{(i)})$:

$$
\frac{\partial L_{\text{a2c}}(\pi^{(i)}, A^{(i)})}{\partial z_y} = \begin{cases} A^{(i)}(\pi^{(i)} - 1), & \text{if } i = y \\ A^{(i)} \pi^{(y)}, & \text{if } i \neq y \end{cases}
\tag{19}
$$

## A.2. Reverse A2C Loss

The derivation of the reverse A2C loss $L_{\mathrm{ra2c}}$ with respect to logits $z$ is presented as follows:

$$L_{\mathrm{ra2c}}(\pi^{(i)}, A^{(i)}) = \begin{cases} \sum_{j \in [k] \setminus \{i\}} -\pi^{(j)} A^{(i)} Z, & \text{if } A^{(i)} > 0 \\ \sum_{j \in [k] \setminus \{i\}} \pi^{(j)} A^{(i)} Z, & \text{if } A^{(i)} < 0 \end{cases} \tag{20}$$

For $i = y$ and $A^{(i)} > 0$,

$$\begin{aligned}
\frac{\partial L_{\mathrm{ra2c}}}{\partial z_y} &= \frac{\partial}{\partial z_y} \sum_{j \in [k] \setminus \{i\}} -\pi^{(j)} A^{(i)} Z \\
&= -A^{(i)} Z \sum_{j \in [k] \setminus \{i\}} \frac{\partial \pi^{(j)}}{\partial z_y} \\
&= -A^{(i)} Z \sum_{j \in [k] \setminus \{i\}} -\pi^{(j)} \pi^{(y)} \quad \text{by (15)} \\
&= A^{(i)} Z \pi^{(y)} (1 - \pi^{(i)}) \\
&= -A^{(i)} Z \pi^{(y)} (\pi^{(y)} - 1)
\end{aligned} \tag{21}$$

For $i \neq y$ and $A^{(i)} > 0$,

$$\begin{aligned}
\frac{\partial L_{\mathrm{ra2c}}}{\partial z_y} &= \frac{\partial}{\partial z_y} \sum_{j \in [k] \setminus \{i\}} -\pi^{(j)} A^{(i)} Z \\
&= -A^{(i)} Z \sum_{j \in [k] \setminus \{i\}} \frac{\partial \pi^{(j)}}{\partial z_y} \\
&= -A^{(i)} Z \left( \sum_{j \in [k]} \frac{\partial \pi^{(j)}}{\partial z_y} - \frac{\partial \pi^{(i)}}{\partial z_y} \right) \\
&= A^{(i)} Z \left( \sum_{j \in [k]} -\pi^{(j)} \pi^{(y)} + \pi^{(y)} \pi^{(y)} + \pi^{(y)} (1 - \pi^{(y)}) - \pi^{(i)} \pi^{(y)} \right) \quad \text{by (14) and (15)} \\
&= -A^{(i)} Z \pi^{(y)} \pi^{(i)}
\end{aligned} \tag{22}$$

For $i = y$ and $A^{(i)} < 0$, the only difference from Equation 21 is the negative sign, thus:

$$\frac{\partial L_{\mathrm{ra2c}}}{\partial z_y} = A^{(i)} Z \pi^{(y)} (\pi^{(y)} - 1) \quad \text{by (21)} \tag{23}$$

For $i \neq y$ and $A^{(i)} < 0$, the only difference from Equation 22 is the negative sign, thus:

$$\frac{\partial L_{\mathrm{ra2c}}}{\partial z_y} = A^{(i)} Z \pi^{(y)} \pi^{(i)} \quad \text{by (22)} \tag{24}$$

In summary, we have the following form for $L_{\mathrm{a2c}}(\pi^{(i)}, A^{(i)})$:

$$\frac{\partial L_{\mathrm{ra2c}}(\pi^{(i)}, A^{(i)})}{\partial z_y} = \begin{cases} -A^{(i)} Z \pi^{(y)} (\pi^{(y)} - 1), & \text{if } i = y \text{ and } A^{(i)} > 0 \\ -A^{(i)} Z \pi^{(y)} \pi^{(i)}, & \text{if } i \neq y \text{ and } A^{(i)} > 0 \\ A^{(i)} Z \pi^{(y)} (\pi^{(y)} - 1), & \text{if } i = y \text{ and } A^{(i)} < 0 \\ A^{(i)} Z \pi^{(y)} \pi^{(i)}, & \text{if } i \neq y \text{ and } A^{(i)} < 0 \end{cases} \tag{25}$$

## A.3. PPO Loss

The derivation of the PPO loss $L_{\text{ppo}}$ with respect to the logits $z$ is presented as follows. The PPO loss includes a clipping function and a minimum operation. When these conditions are not satisfied, there is no gradient.

The sample-wise PPO loss is:

$$L_{\text{ppo}}(\pi^{(i)}, A^{(i)}, \pi_{\text{old}}^{(i)}) = -\frac{\pi^{(i)}}{\pi_{\text{old}}^{(i)}} A^{(i)} \tag{26}$$

For $i = y$,

$$
\begin{aligned}
\frac{\partial L_{\text{ppo}}}{\partial z_y} &= \frac{\partial}{\partial z_y} - \frac{\pi^{(i)}}{\pi_{\text{old}}^{(i)}} A^{(i)} \\
&= -\frac{A^{(i)}}{\pi_{\text{old}}^{(i)}} \frac{\partial \pi^{(i)}}{\partial z_y} \\
&= \frac{A^{(i)} \pi^{(i)} (\pi^{(i)} - 1)}{\pi_{\text{old}}^{(i)}} \quad \text{by (14)}
\end{aligned}
\tag{27}
$$

For $i \neq y$,

$$
\begin{aligned}
\frac{\partial L_{\text{ppo}}}{\partial z_y} &= \frac{\partial}{\partial z_y} - \frac{\pi^{(i)}}{\pi_{\text{old}}^{(i)}} A^{(i)} \\
&= \frac{A^{(i)}}{\pi_{\text{old}}^{(i)}} \frac{\partial \pi^{(i)}}{\partial z_y} \\
&= \frac{A^{(i)} \pi^{(i)} \pi^{(y)}}{\pi_{\text{old}}^{(i)}} \quad \text{by (15)}
\end{aligned}
\tag{28}
$$

## A.4. Reverse PPO Loss

The derivation of the reverse PPO loss $L_{\text{rppo}}$ with respect to logits $z$ is presented as follows. As with PPO, the reverse PPO loss only considers samples that pass the clipping function and the minimum operation.

From Section A.2, we have the following form for $L_{\text{rppo}}(\pi^{(i)}, A^{(i)}, \pi_{\text{old}}^{(i)})$:

$$
\frac{\partial L_{\text{rppo}}(\pi^{(i)}, A^{(i)}, \pi_{\text{old}}^{(i)})}{\partial z_y} = \begin{cases}
-\frac{A^{(i)} Z \pi^{(y)} (\pi^{(y)} - 1)}{\pi_{\text{old}}^{(i)}}, & \text{if } i = y \text{ and } A^{(i)} > 0 \\
-\frac{A^{(i)} Z \pi^{(y)} \pi^{(i)}}{\pi_{\text{old}}^{(i)}}, & \text{if } i \neq y \text{ and } A^{(i)} > 0 \\
\frac{A^{(i)} Z \pi^{(y)} (\pi^{(y)} - 1)}{\pi_{\text{old}}^{(i)}}, & \text{if } i = y \text{ and } A^{(i)} < 0 \\
\frac{A^{(i)} Z \pi^{(y)} \pi^{(i)}}{\pi_{\text{old}}^{(i)}}, & \text{if } i \neq y \text{ and } A^{(i)} < 0
\end{cases}
\tag{29}
$$

# B. Gradient Analysis of RL Loss and Reverse RL Loss

## B.1. Symmetric A2C Gradient Analysis

The gradient analysis of the symmetric RL loss follows the SCE analysis. We adopt their analysis and extend it to cover the RL loss analysis. We set $\alpha$ and $\beta$ to 1 for simplicity and evaluate the gradient direction of both RL and reverse RL losses with respect to the logits $z$. We show that the gradient directions for both types are the same and that the reverse RL loss helps deviate ambiguous predictions where the probability is around 0.5. We first show how the symmetric A2C (SA2C) loss behaves. Note that $Z < 0$ is a constant used in the reverse RL loss to handle $\log 0 = -\infty$.

$$L_{\text{sa2c}} = L_{\text{a2c}} + L_{\text{ra2c}} \tag{30}$$

For $i = y$ and $A^{(i)} > 0$,

$$
\begin{aligned}
\frac{\partial L_{\text{sa2c}}}{\partial z_y} &= \frac{\partial L_{\text{a2c}}}{\partial z_y} + \frac{\partial L_{\text{ra2c}}}{\partial z_y} \\
&= \underbrace{A^{(i)}(\pi^{(i)} - 1)}_{\nabla L_{\text{a2c}} < 0} \underbrace{-A^{(i)} Z \pi^{(i)}(\pi^{(i)} - 1)}_{\nabla L_{\text{ra2c}} < 0} \quad \text{by (17) and (21)}
\end{aligned}
\tag{31}
$$

For $i \neq y$ and $A^{(i)} > 0$,

$$
\begin{aligned}
\frac{\partial L_{\text{sa2c}}}{\partial z_y} &= \frac{\partial L_{\text{a2c}}}{\partial z_y} + \frac{\partial L_{\text{ra2c}}}{\partial z_y} \\
&= \underbrace{A^{(i)}\pi^{(y)}}_{\nabla L_{\text{a2c}} > 0} \underbrace{-A^{(i)} Z \pi^{(y)}\pi^{(i)}}_{\nabla L_{\text{ra2c}} > 0} \quad \text{by (18) and (22)}
\end{aligned}
\tag{32}
$$

For $i = y$ and $A^{(i)} < 0$,

$$
\begin{aligned}
\frac{\partial L_{\text{sa2c}}}{\partial z_y} &= \frac{\partial L_{\text{a2c}}}{\partial z_y} + \frac{\partial L_{\text{ra2c}}}{\partial z_y} \\
&= \underbrace{A^{(i)}(\pi^{(i)} - 1)}_{\nabla L_{\text{a2c}} > 0} \underbrace{-A^{(i)} Z \pi^{(y)}(\pi^{(y)} - 1)}_{\nabla L_{\text{ra2c}} > 0} \quad \text{by (17) and (21)}
\end{aligned}
\tag{33}
$$

For $i \neq y$ and $A^{(i)} < 0$,

$$
\begin{aligned}
\frac{\partial L_{\text{sa2c}}}{\partial z_y} &= \frac{\partial L_{\text{a2c}}}{\partial z_y} + \frac{\partial L_{\text{ra2c}}}{\partial z_y} \\
&= \underbrace{A^{(i)}\pi^{(y)}}_{\nabla L_{\text{a2c}} < 0} \underbrace{-A^{(i)} Z \pi^{(y)}\pi^{(i)}}_{\nabla L_{\text{ra2c}} < 0} \quad \text{by (18) and (22)}
\end{aligned}
\tag{34}
$$

For the above cases, the gradient directions of the RL (A2C) loss and the reverse RL (RA2C) loss are the same as SCE gradients. Essentially, the RA2C loss acts as an accelerator. In the case of $i = y$ and $A^{(i)} > 0$, the gradient of the RA2C loss is $-A^{(i)} Z \pi^{(y)}(\pi^{(y)} - 1)$, with the largest gradient magnitude at $\pi^{(y)} = 0.5$ as a parabolic function. In other words, the accelerator helps the probability $\pi^{(i)}$ increase most quickly when it is ambiguous which action to take. In the case of $i \neq y$ and $A^{(i)} > 0$, the probability of other actions except $a^{(i)}$ is reduced, and this reduction is influenced by the confidence of both $\pi^{(i)}$ and $\pi^{(y)}$. Specifically, the gradient of the RA2C loss is $-A^{(i)} Z \pi^{(y)}\pi^{(i)}$. When both $\pi^{(i)}$ and $\pi^{(y)}$ are 0.5, indicating the most ambiguous predictions, the accelerator helps the A2C loss reduce $\pi^{(y)}$ most aggressively.

When $A^{(i)} < 0$, the gradient direction is simply reversed. The behavior of the gradient itself remains the same as when $A^{(i)} > 0$. In the case of $i = y$, RA2C decreases the probability $\pi^{(y)}$ more when $\pi^{(y)}$ is around 0.5. For $i \neq y$, RA2C helps increase $\pi^{(y)}$ more when both $\pi^{(i)}$ and $\pi^{(y)}$ are ambiguous (both around 0.5).

### B.2. Symmetric PPO Gradient Analysis

$$
L_{\text{sppo}} = L_{\text{ppo}} + L_{\text{rppo}}
\tag{35}
$$

For $i = y$ and $A^{(i)} > 0$,

$$
\begin{aligned}
\frac{\partial L_{\text{sppo}}}{\partial z_y} &= \frac{\partial L_{\text{ppo}}}{\partial z_y} + \frac{\partial L_{\text{rppo}}}{\partial z_y} \\
&= \underbrace{\frac{A^{(i)}\pi^{(i)}(\pi^{(i)} - 1)}{\pi_{\text{old}}^{(i)}}}_{\nabla L_{\text{ppo}} < 0} - \underbrace{\frac{A^{(i)} Z \pi^{(y)}(\pi^{(y)} - 1)}{\pi_{\text{old}}^{(i)}}}_{\nabla L_{\text{rppo}} < 0} \quad \text{by (27) and (29)}
\end{aligned}
\tag{36}
$$

For $i \neq y$ and $A^{(i)} > 0$,

$$
\begin{aligned}
\frac{\partial L_{\text{sppo}}}{\partial z_y} &= \frac{\partial L_{\text{ppo}}}{\partial z_y} + \frac{\partial L_{\text{rppo}}}{\partial z_y} \\
&= \underbrace{\frac{A^{(i)}\pi^{(i)}\pi^{(y)}}{\pi_{\text{old}}^{(i)}}}_{\nabla L_{\text{ppo}} > 0} - \underbrace{\frac{A^{(i)}Z\pi^{(y)}\pi^{(i)}}{\pi_{\text{old}}^{(i)}}}_{\nabla L_{\text{rppo}} > 0} \quad \text{by (28) and (29)}
\end{aligned}
\tag{37}
$$

For $i = y$ and $A^{(i)} < 0$,

$$
\begin{aligned}
\frac{\partial L_{\text{sppo}}}{\partial z_y} &= \frac{\partial L_{\text{ppo}}}{\partial z_y} + \frac{\partial L_{\text{rppo}}}{\partial z_y} \\
&= \underbrace{\frac{A^{(i)}\pi^{(i)}\left(\pi^{(i)} - 1\right)}{\pi_{\text{old}}^{(i)}}}_{\nabla L_{\text{ppo}} < 0} + \underbrace{\frac{A^{(i)}Z\pi^{(y)}\left(\pi^{(y)} - 1\right)}{\pi_{\text{old}}^{(i)}}}_{\nabla L_{\text{rppo}} < 0} \quad \text{by (27) and (29)}
\end{aligned}
\tag{38}
$$

For $i \neq y$ and $A^{(i)} < 0$,

$$
\begin{aligned}
\frac{\partial L_{\text{sppo}}}{\partial z_y} &= \frac{\partial L_{\text{ppo}}}{\partial z_y} + \frac{\partial L_{\text{rppo}}}{\partial z_y} \\
&= \underbrace{\frac{A^{(i)}\pi^{(i)}\pi^{(y)}}{\pi_{\text{old}}^{(i)}}}_{\nabla L_{\text{ppo}} > 0} + \underbrace{\frac{A^{(i)}Z\pi^{(y)}\pi^{(i)}}{\pi_{\text{old}}^{(i)}}}_{\nabla L_{\text{rppo}} > 0} \quad \text{by (28) and (29)}
\end{aligned}
\tag{39}
$$

Basically, the mechanism of RPPO is the same as RA2C, except for $\pi_{\text{old}}^{(i)}$, which does not change the gradient sign. Therefore, RPPO also helps PPO deviate from ambiguous predictions, acting as an accelerator.

## C. Experimental Setups and Results

### C.1. Hyperparameters

**Atari Games:** We primarily follow the hyperparameter settings of RL Baselines3 Zoo (Raffin, 2020). Most hyperparameter values remain unchanged across environments. Only $\alpha$ and $\beta$ are adjusted for the reverse RL loss. For SA2C without noise, we use $(\alpha = 0.5, \beta = 5.0)$ for all environments. For SA2C with noise, we use $(\alpha = 0.5, \beta = 1.0)$ for (Alien, MsPacman, Qbert, TimePilot, VideoPinball, Assault, Gravitar, StarGunner, UpNDown), and $(\alpha = 0.5, \beta = 1.0)$ for others. For SPPO without noise, we use $(\alpha = 0.5, \beta = 1.0)$ for all environments. For SPPO with noise, we use $(\alpha = 0.5, \beta = 10.0)$ for all environments. We do not use any GPU for Atari games.

*Table 4.* Hyperparameters for Atari games

|  | **WITHOUT NOISE** | $\epsilon \sim \mathbf{BSC}(0.1)$ |
|---|---|---|
| SA2C |  |  |
| - $(\alpha = 0.5, \beta = 1.0)$ | - | (ALIEN, ASSAULT, GRAVITAR, MSPACMAN, QBERT, STARGUNNER, TIMEPILOT, UPNDOWN, VIDEOPINBALL) |
| - $(\alpha = 0.5, \beta = 5.0)$ | ALL ENVIRONMENTS | ALL OTHERS EXCEPT THOSE MENTIONED ABOVE |
| SPPO |  |  |
| - $(\alpha = 0.5, \beta = 1.0)$ | ALL ENVIRONMENTS | - |
| - $(\alpha = 0.5, \beta = 10.0)$ | - | ALL ENVIRONMENTS |

**MuJoCo and Box2D:** We use n_envs = 4 and n_steps = 8 for A2C and SA2C. We follow Stable-Baselines3's default hyperparameters (Raffin et al., 2021) for other settings. Only $\alpha$ and $\beta$ are adjusted for the reverse RL loss. For table visibility, let {Ant = 1, BipedalWalker = 2, HalfCheetah = 3, Hopper = 4, HumanoidStandup = 5, InvertedDoublePendulum = 6, LunarLanderContinuous = 7, Swimmer = 8, Walker2d = 9}. We do not use any GPU for these tasks.

*Table 5.* Hyperparameters for MuJoCO and Box2D environments

|  | **WITHOUT NOISE** | $\epsilon \sim \mathcal{N}(0, 0.05^2)$ |
|---|---|---|
| SA2C |  |  |
| - $(\alpha = 0.5, \beta = 0.2)$ | (1, 4, 8, 9) | (1) |
| - $(\alpha = 0.5, \beta = 0.5)$ | (2, 5) | - |
| - $(\alpha = 0.5, \beta = 5.0)$ | (3, 6, 7) | (7) |
| - $(\alpha = 0.5, \beta = 10.0)$ | - | (2, 3, 4, 5, 6, 8, 9) |
| - $Z = -1$ | ALL ENVIRONMENTS | ALL ENVIRONMENTS |
| - (TIMESTEPS= 2e6) | ALL ENVIRONMENTS | ALL ENVIRONMENTS |
| - (NUMBER OF BINS= 11) | ALL ENVIRONMENTS | ALL ENVIRONMENTS |
| SPPO |  |  |
| - $(\alpha = 0.5, \beta = 20.0)$ | ALL ENVIRONMENTS | (1, 7) |
| - $(\alpha = 0.5, \beta = 25.0)$ | - | (2, 3, 5, 6, 9) |
| - $(\alpha = 0.5, \beta = 50.0)$ | - | (4, 8) |
| - $Z = -1$ | ALL ENVIRONMENTS | ALL ENVIRONMENTS |
| - (TIMESTEPS= 1e6) | (2, 6) | (2, 6) |
| - (TIMESTEPS= 2e6) | (1, 3, 4, 8, 9) | (1, 3, 4, 8, 9) |
| - (TIMESTEPS= 5e6) | (9) | (9) |
| - (TIMESTEPS= 1e7) | (5) | (5) |
| - (NUMBER OF BINS= 11) | ALL ENVIRONMENTS | ALL ENVIRONMENTS |

**IMDB and TL;DR:** We basically use the provided implementation (Chang et al., 2023) and follow their hyperparameters, with the addition of the advantage normalization step for PPO. The scripts used in our experiments are available in the code repository for further detail. We use a single Nvidia A100 (80GB) for our experiments.

*Table 6.* Hyperparameters for IMDB positive sentiment and TL;DR summarization

|  | **IMDB** | **TL;DR** |
|---|---|---|
| PPO |  |  |
| - MODEL: | GPT-2 | GPT-J |
| - UPDATES: | 60 | 100 |
| - TRAJECTORIES PER UPDATE: | 112 | 64 |
| - EPOCHS PER UPDATE: | 5 | 4 |
| - BATCH SIZE | 28 | 32 |
| - LEARNING RATE | 5E-6 | 5E-6 |
| - DISCOUNT FACTOR | 0.99 | 1.0 |
| - GAE LAMBDA | 0.95 | 0.95 |
| - CLIP RANGE | 0.2 | 0.2 |
| SPPO | $(\alpha = 0.5, \beta = 0.4)$ | $(\alpha = 0.5, \beta = 0.2)$ |

## C.2. Experimental Results: A2C and SA2C

*Table 7.* Mean final scores and standard errors (over the last 10 episodes) of A2C and SA2C on Atari games, without and with binary symmetric channel (BSC) noise with a crossover probability of 0.1 across 5 seeds.

|  | **WITHOUT NOISE** |  | $\epsilon \sim$ **BSC**$(0.1)$ |  |
|---|---|---|---|---|
|  | A2C | SA2C | A2C | SA2C |
| ALIEN | $\mathbf{913 \pm 100}$ | $771 \pm 51$ | $481 \pm 72$ | $\mathbf{496 \pm 37}$ |
| ASSAULT | $\mathbf{1538 \pm 199}$ | $1061 \pm 41$ | $287 \pm 226$ | $\mathbf{399 \pm 133}$ |
| ASTERIX | $2308 \pm 86$ | $\mathbf{2377 \pm 164}$ | $1403 \pm 305$ | $\mathbf{1430 \pm 208}$ |
| BEAMRIDER | $1121 \pm 61$ | $\mathbf{1335 \pm 43}$ | $\mathbf{1087 \pm 339}$ | $902 \pm 196$ |
| CENTIPEDE | $\mathbf{3588 \pm 430}$ | $3574 \pm 295$ | $3108 \pm 243$ | $\mathbf{3540 \pm 194}$ |
| CRAZYCLIMBER | $98774 \pm 2516$ | $\mathbf{99330 \pm 4371}$ | $93042 \pm 8711$ | $\mathbf{97058 \pm 6251}$ |
| DEMONATTACK | $4309 \pm 325$ | $\mathbf{5017 \pm 625}$ | $\mathbf{30 \pm 21}$ | $19 \pm 3$ |
| FROSTBITE | $255 \pm 2$ | $\mathbf{257 \pm 3}$ | $241 \pm 9$ | $\mathbf{286 \pm 48}$ |
| GOPHER | $960 \pm 80$ | $\mathbf{1036 \pm 138}$ | $947 \pm 91$ | $\mathbf{996 \pm 114}$ |
| GRAVITAR | $143 \pm 18$ | $\mathbf{201 \pm 16}$ | $\mathbf{279 \pm 48}$ | $183 \pm 36$ |
| KRULL | $6387 \pm 267$ | $\mathbf{7672 \pm 819}$ | $\mathbf{7564 \pm 486}$ | $6337 \pm 754$ |
| MSPACMAN | $1175 \pm 43$ | $\mathbf{1495 \pm 104}$ | $\mathbf{926 \pm 44}$ | $916 \pm 100$ |
| NAMETHISGAME | $\mathbf{5945 \pm 102}$ | $5614 \pm 166$ | $2280 \pm 257$ | $\mathbf{2372 \pm 141}$ |
| QBERT | $1646 \pm 240$ | $\mathbf{2103 \pm 261}$ | $620 \pm 96$ | $\mathbf{641 \pm 77}$ |
| RIVERRAID | $4368 \pm 582$ | $\mathbf{5461 \pm 456}$ | $1609 \pm 65$ | $\mathbf{2511 \pm 190}$ |
| ROADRUNNER | $14971 \pm 1396$ | $\mathbf{18624 \pm 1812}$ | $\mathbf{5606 \pm 1788}$ | $3830 \pm 1517$ |
| SEAQUEST | $836 \pm 7$ | $\mathbf{988 \pm 92}$ | $650 \pm 22$ | $\mathbf{653 \pm 22}$ |
| STARGUNNER | $\mathbf{2222 \pm 114}$ | $1766 \pm 120$ | $\mathbf{1194 \pm 645}$ | $622 \pm 54$ |
| TIMEPILOT | $\mathbf{3992 \pm 198}$ | $3116 \pm 137$ | $2232 \pm 259$ | $\mathbf{3288 \pm 106}$ |
| UPNDOWN | $\mathbf{8313 \pm 1544}$ | $1638 \pm 761$ | $4228 \pm 1187$ | $\mathbf{7093 \pm 2772}$ |
| VIDEOPINBALL | $\mathbf{24948 \pm 3038}$ | $19618 \pm 1888$ | $20319 \pm 2157$ | $\mathbf{25035 \pm 3914}$ |
| WIZARDOFWOR | $\mathbf{824 \pm 136}$ | $674 \pm 125$ | $496 \pm 87$ | $\mathbf{752 \pm 156}$ |
| **WINS (SA2C)** | **12 / 22** |  | **15 / 22** |  |

*Table 8.* Mean final scores and standard errors (over the last 10 episodes) of A2C and SA2C on MuJoCo benchmark tasks and Box2D environments without Gaussian noise across 30 seeds.

| WITHOUT NOISE | ANT | HOPPER | HALFCHEETAH | HUMANOIDSTANDUP |
|---|---|---|---|---|
| A2C | $757 \pm 116$ | $1410 \pm 112$ | $1393 \pm 163$ | $121850 \pm 4264$ |
| DA2C | $2220 \pm 96$ | $\mathbf{1944 \pm 116}$ | $\mathbf{2325 \pm 209}$ | $152135 \pm 3937$ |
| **DSA2C** | $\mathbf{2287 \pm 94}$ | $1797 \pm 139$ | $2266 \pm 203$ | $\mathbf{159142 \pm 129}$ |
| | WALKER2D | SWIMMER | BIPEDALWALKER | LUNARLANDERCONTINUOUS |
| A2C | $1348 \pm 130$ | $95.8 \pm 19.0$ | $124 \pm 23$ | $79.0 \pm 20.2$ |
| DA2C | $\mathbf{2131 \pm 154}$ | $\mathbf{142.4 \pm 17.0}$ | $234 \pm 22$ | $176.7 \pm 20.9$ |
| **DSA2C** | $1662 \pm 164$ | $128.5 \pm 16.2$ | $\mathbf{274 \pm 16}$ | $\mathbf{221.2 \pm 10.7}$ |
| | INVERTEDDOUBLEPENDULUM | | | |
| A2C | $1670 \pm 500$ | | | |
| DA2C | $9139 \pm 94$ | | | |
| **DSA2C** | $\mathbf{9145 \pm 93}$ | | | |

*Table 9.* Mean final scores and standard errors (over the last 10 episodes) of A2C and SA2C on MuJoCo benchmark tasks and Box2D environments with Gaussian noise (mean 0 and standard deviation 0.05) across 30 seeds.

| $\epsilon \sim \mathcal{N}(0, 0.05^2)$ | ANT | HOPPER | HALFCHEETAH | HUMANOIDSTANDUP |
|---|---|---|---|---|
| A2C | $673 \pm 108$ | $1083 \pm 92$ | $1610 \pm 163$ | $101064 \pm 4933$ |
| DA2C | $1296 \pm 80$ | $\mathbf{1323 \pm 87}$ | $1510 \pm 126$ | $126241 \pm 3973$ |
| **DSA2C** | $\mathbf{1520 \pm 83}$ | $1307 \pm 102$ | $\mathbf{1696 \pm 163}$ | $\mathbf{128064 \pm 4391}$ |
| | WALKER2D | SWIMMER | BIPEDALWALKER | LUNARLANDERCONTINUOUS |
| A2C | $786 \pm 86$ | $28.9 \pm 4.4$ | $158 \pm 20$ | $-3.7 \pm 15.9$ |
| DA2C | $\mathbf{1599 \pm 138}$ | $36.8 \pm 4.6$ | $210 \pm 21$ | $106 \pm 20$ |
| **DSA2C** | $1423 \pm 129$ | $\mathbf{53.1 \pm 7.0}$ | $\mathbf{222 \pm 20}$ | $\mathbf{179 \pm 12}$ |
| | INVERTEDDOUBLEPENDULUM | | | |
| A2C | $3852 \pm 634$ | | | |
| DA2C | $7900 \pm 364$ | | | |
| **DSA2C** | $\mathbf{8323 \pm 217}$ | | | |

## C.3. Experimental Results: PPO and SPPO

*Table 10.* Mean final scores and standard errors (over the last 10 episodes) of PPO and SPPO on Atari games, without and with binary symmetric channel (BSC) noise with a crossover probability of 0.1 across 5 seeds.

| | **WITHOUT NOISE** | | $\epsilon \sim$ **BSC**$(0.1)$ | |
| | PPO | **SPPO** | PPO | **SPPO** |
|---|---|---|---|---|
| ALIEN | **1128 ± 105** | 1081 ± 79 | 525 ± 26 | **713 ± 26** |
| ASSAULT | 3134 ± 193 | **3385 ± 214** | 2327 ± 401 | **3698 ± 363** |
| ASTERIX | 2599 ± 101 | **2976 ± 150** | 1272 ± 106 | **1739 ± 329** |
| BEAMRIDER | **2176 ± 251** | 1635 ± 404 | **1828 ± 130** | 1580 ± 96 |
| CENTIPEDE | 2961 ± 379 | **3694 ± 224** | 4759 ± 257 | **7525 ± 769** |
| CRAZYCLIMBER | 86764 ± 3568 | **103588 ± 2871** | 71144 ± 11060 | **99810 ± 2487** |
| DEMONATTACK | 7872 ± 302 | **7901 ± 455** | **161 ± 24** | 132 ± 13 |
| FROSTBITE | 268 ± 5 | **286 ± 6** | **509 ± 108** | 23 ± 16 |
| GOPHER | 787 ± 48 | **875 ± 78** | 478 ± 38 | **7765 ± 3366** |
| GRAVITAR | 371 ± 47 | **442 ± 67** | 269 ± 39 | **332 ± 61** |
| KRULL | 6628 ± 417 | **7578 ± 588** | 5602 ± 481 | **9015 ± 381** |
| MSPACMAN | 837 ± 62 | **1204 ± 86** | 704 ± 41 | **1011 ± 52** |
| NAMETHISGAME | **5665 ± 280** | 5423 ± 63 | 2681 ± 143 | **5187 ± 247** |
| QBERT | 4352 ± 128 | **4412 ± 282** | 2827 ± 1927 | **4020 ± 2415** |
| RIVERRAID | 6128 ± 272 | **6343 ± 219** | 2460 ± 127 | **3998 ± 248** |
| ROADRUNNER | **28382 ± 2254** | 22562 ± 2875 | 1204 ± 157 | **3830 ± 1230** |
| SEAQUEST | **902 ± 2** | 888 ± 6 | 652 ± 16 | **814 ± 15** |
| STARGUNNER | 11848 ± 722 | **14746 ± 1876** | 1514 ± 110 | **23250 ± 6292** |
| TIMEPILOT | **3850 ± 151** | 3548 ± 220 | 3506 ± 318 | **3936 ± 420** |
| UPNDOWN | 58289 ± 21226 | **126830 ± 27534** | 8815 ± 1395 | **73490 ± 33553** |
| VIDEOPINBALL | 22408 ± 4292 | **29485 ± 2851** | 31680 ± 2318 | **37048 ± 6989** |
| WIZARDOFWOR | 3186 ± 256 | **3762 ± 387** | 940 ± 158 | **4442 ± 1332** |
| **WINS (SPPO)** | **16 / 22** | | **19 / 22** | |

*Table 11.* Percentage improvement of SPPO over PPO. The percentage improvements are computed across 22 Atari games. We simply fix $\alpha = 0.5$ to reduce the total loss magnitude and vary $\beta$ to control the impact of the reverse RL loss. We exclude very large improvements (e.g., 2000%) from calculating the average. This large improvements result from PPO's significant learning failures.

| $\alpha = 0.5$ IS FIXED | $\beta = 0.5$ | $\beta = 1.0$ | $\beta = 5.0$ | $\beta = 10.0$ | $\beta = 25.0$ |
|---|---|---|---|---|---|
| SPPO UNDER 0% NOISE | 7.83% | 10.15% | 24.98% | 21.52% | 18.92% |
| SPPO UNDER 10% NOISE | 1.74% | 21.89% | 148.46% | 166.73% | 136.50% |

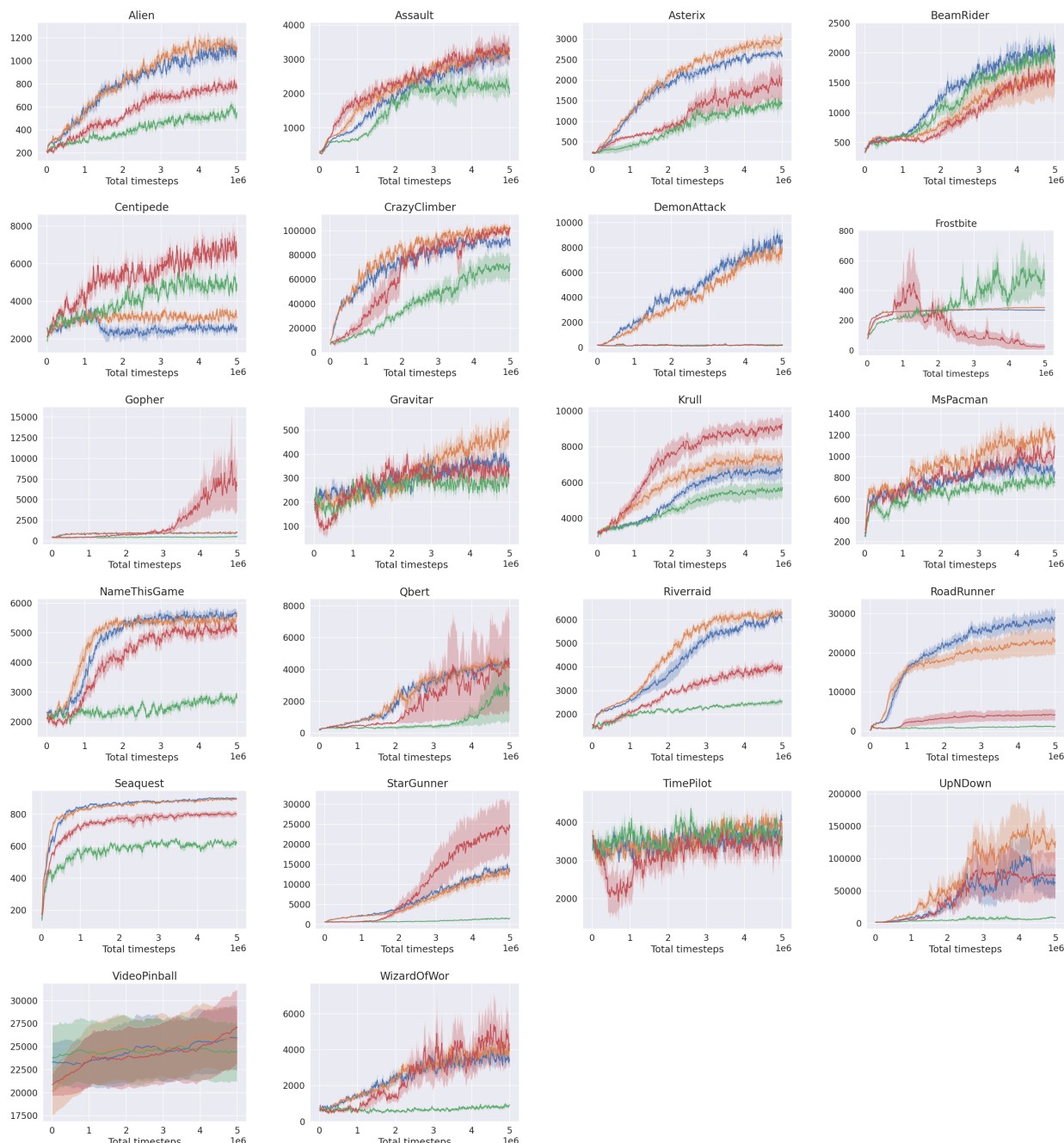

*Figure 4.* Result of training plots for SPPO and PPO for Atari games. The blue line indicates the original PPO without any added noise, while the orange line represents SPPO without added noise. The green line indicates PPO with 10% noise, and the red line represents SPPO with 10% noise. We fix $\alpha = 0.5$ for all environments, with $\beta = 1.0$ for the experiments without noise and $\beta = 10.0$ for the noise environments.

*Table 12.* Mean final scores and standard errors (over the last 10 episodes) of PPO and SPPO on MuJoCo benchmark tasks and Box2D environments without Gaussian noise across 30 seeds.

| WITHOUT NOISE | ANT | HOPPER | HALFCHEETAH | HUMANOIDSTANDUP |
|---|---|---|---|---|
| PPO | $2068 \pm 166$ | $\mathbf{2875 \pm 137}$ | $2282 \pm 191$ | $93763 \pm 3402$ |
| DPPO | $2735 \pm 109$ | $2154 \pm 119$ | $3478 \pm 279$ | $176320 \pm 6538$ |
| **DSPPO** | $\mathbf{2885 \pm 100}$ | $2299 \pm 115$ | $\mathbf{4104 \pm 258}$ | $\mathbf{189301 \pm 5915}$ |
| | WALKER2D | SWIMMER | BIPEDALWALKER | LUNARLANDERCONTINUOUS |
| PPO | $2793 \pm 199$ | $112 \pm 5.0$ | $247 \pm 8.3$ | $134 \pm 10.9$ |
| DPPO | $4443 \pm 119$ | $\mathbf{131 \pm 0.3}$ | $265 \pm 15.5$ | $241 \pm 7.7$ |
| **DSPPO** | $\mathbf{4587 \pm 154}$ | $130 \pm 0.6$ | $\mathbf{274 \pm 6.2}$ | $\mathbf{250 \pm 6.9}$ |
| | INVERTEDDOUBLEPENDULUM | | | |
| PPO | $7454 \pm 394$ | | | |
| DPPO | $8928 \pm 136$ | | | |
| **DSPPO** | $\mathbf{9015 \pm 101}$ | | | |

*Table 13.* Mean final scores and standard errors (over the last 10 episodes) of PPO and SPPO on MuJoCo benchmark tasks and Box2D environments with Gaussian noise (mean 0 and standard deviation 0.05) across 30 seeds.

| $\epsilon \sim \mathcal{N}(0, 0.05^2)$ | ANT | HOPPER | HALFCHEETAH | HUMANOIDSTANDUP |
|---|---|---|---|---|
| PPO | $601 \pm 47$ | $1936 \pm 147$ | $2068 \pm 208$ | $80945 \pm 2130$ |
| DPPO | $1897 \pm 86$ | $2153 \pm 106$ | $2722 \pm 188$ | $\mathbf{146038 \pm 1841}$ |
| **DSPPO** | $\mathbf{2095 \pm 102}$ | $\mathbf{2333 \pm 109}$ | $\mathbf{3118 \pm 195}$ | $145974 \pm 2520$ |
| | WALKER2D | SWIMMER | BIPEDALWALKER | LUNARLANDERCONTINUOUS |
| PPO | $1270 \pm 107$ | $44 \pm 3.0$ | $158 \pm 15.2$ | $181 \pm 13.8$ |
| DPPO | $3419 \pm 100$ | $57 \pm 3.6$ | $\mathbf{274 \pm 7.1}$ | $281 \pm 5.7$ |
| **DSPPO** | $\mathbf{3523 \pm 129}$ | $\mathbf{72 \pm 5.1}$ | $267 \pm 8.8$ | $\mathbf{294 \pm 3.3}$ |
| | INVERTEDDOUBLEPENDULUM | | | |
| PPO | $8050 \pm 244$ | | | |
| DPPO | $8963 \pm 100$ | | | |
| **DSPPO** | $\mathbf{9147 \pm 61}$ | | | |

## C.4. On and Off Advantage Normalization

*Table 14.* Comparison with and without advantage normalization over 4 different random seeds.

| PPO | IMDB | TL;DR |
|---|---|---|
| WITHOUT $A$ NORMALIZATION | $0.77 \pm 0.01$ | $6.06 \pm 0.02$ |
| WITH $A$ NORMALIZATION | $0.89 \pm 0.02$ | $5.94 \pm 0.08$ |

# D. Examples of Reward Model Errors

## Warning: This section contains harmful language.

*Table 15.* Example showing a trained reward model with errors that are not consistent for empty outputs, and the reward for an empty output is greater than that for a non-empty summarization. [...] indicates omitted content for brevity.

---

**SUBREDDIT:** r/RELATIONSHIPS (SAMPLE ID: 37)
**TITLE:** I'M A DUMB [21] MALE AND SO I'M HAVING A LOT OF TROUBLE INTERPRETING THE SIGNALS THAT THIS [21] GIRL MAY OR MAY NOT BE SENDING ME. A LITTLE HELP PLEASE?
**POST:** SO OKAY, I'M FROM NEW YORK BUT I STUDY IN OREGON FOR MOST OF THE YEAR. RECENTLY A FRIEND OF MINE WHO I WAS NOT REALLY CLOSE STARTED FACEBOOK MESSAGING ME, THAT WAS ABOUT 3 MONTHS AGO, SINCE THEN WE'VE TALKED ALMOST EVERYDAY. […] I TRIED TO DO JUST THAT BUT SHE TOTALLY GAVE ME THE COLD SHOULDER; NOT BEING REALLY RESPONSIVE TO HANGING OUT, LEAVING EARLY WHEN WE FINALLY DID ETC... AM I WRONG IN MY ORIGINAL ASSUMPTION THAT SHE WAS INTO ME JUST BECAUSE OUT OF THE BLUE SHE STARTED TALKING TO ME A LOT? IS SHE TRYING TO PLAY HARD TO GET? AM I LOOKING WAY TOO INTO THIS AND MAYBE SHE WAS JUST OCCUPIED THAT WEEKEND? I REALLY HAVE NO IDEA HOW TO EVALUATE THIS. DO ANY OF YOU GUYS HAVE ANY SUGGESTIONS/IDEAS?

**GENERATED SUMMARY:** <EMPTY>

**REWARD MODEL OUTPUT:** 6.66

---

**SUBREDDIT:** r/RELATIONSHIP_ADVICE (SAMPLE ID: 60)
**TITLE:** MY BF [23] DOESN'T SPEAK OF HIS CHILDHOOD, BUT I[F22] KNOW HE'S TRAUMATIZED.
**POST:** WE WERE FRIENDS FOR 10 YEARS, BEFORE WE GOT TOGETHER. HE THAN TOLD ME ONCE ABOUT HIS TERRIBLE CHILDHOOD. (HE TOLD ONLY 3 OF HIS FRIENDS HIS STORY) NOW WE'RE A COUPLE FOR QUITE A FEW MONTHS AND WELL, SOMETIMES THERE'S STUFF I KNOW THAT REMINDS HIM OF HIS CHILDHOOD, BUT IT'S LIKE HE'S FORGOTTEN THAT HE HAD TOLD ME. […] AND STUFF LIKE WATCHING TVSHOWS ABOUT RAISING CHILDREN. WE TALK ABOUT HOW WE'RE GOING TO RAISE OURS IN THE FUTURE AND THAT WE WON'T WILL BE AS HORRIBLE AS THE PARENTS ON TV. (BUT STRIKING, THE THINGS HE THINKS ARE IMPORTANT ARE ALWAYS THE THINGS HIS PARENTS SHOULD HAVE DONE, TO SAVE HIM FROM THE TRAUMATIZING STUFF.)I KNOW HE LIKES TO PUT HIS PROBLEMS FAR AWAY. BUT ON THE OTHER HAND, I'M HIS GIRLFRIEND NOW AND WE'RE PRETTY SERIOUS, ISN'T IT GOOD TO SPEAK ABOUT IT MAYBE JUST ONCE, SO HE KNOWS I KNOW HIS SECRET/WON'T TELL, AND MOST OF ALL, I'M ALWAYS THERE FOR HIM? WHAT DO YOU THINK?

**GENERATED SUMMARY:** <EMPTY>

**REWARD MODEL OUTPUT:** 3.14

---

**SUBREDDIT:** r/ASKREDDIT (SAMPLE ID: 27)
**TITLE:** DEAR REDDIT, WHAT SILLY/IRRELEVANT/REDICULOUS FAMILY MISCOMMUNICATIONS HAVE LEAD TO FEUDS LASTING YEARS?
**POST:** MY GRANDMA AND MY AUNT (HER DAUGHER-IN-AW) HAVEN'T SPOKEN TO EACH OTHER IN YEARS OVER A PHONE THAT DIDN'T GET HUNG UP. MY AUNT AND UNCLE SCREEN THEIR CALLS AND FREQUENTLY DO NOT RETURN THEM– ONE TIME, MY GRANDMA CALLED AND LEFT A MESSAGE THEN THOUGHT SHE HUNG UP THE PHONE. A FEW MINUTES LATER– MY GRANDMA WAS TALKING WITH SOMEONE IN HER HOME AND USED THE WORD "BITCH"- THIS WAS ALL RECORDED ON MY AUNT AND UNCLE'S ANSWERING MACHINE AND MY AUNT ASSUMED IT WAS ABOUT HER AND HASN'T SPOKEN TO NOR SEEN MY GRANDMA IN UPWARDS OF 5 YEARS. […] WHY WASTE TIME THE TIME YOU HAVE WITH SOMONE? WHY CONTINUE TO HOLD A SILLY GRUDGE? TO COMPLICATE MATTERS FURTHER, MY GRANDMA HAS A DAUGHTER WHO LIVES WITH HER AND LIKES TO BE IN OTHER PEOPLES BUSINESS– I THINK SHE IS ALSO PART OF THE PROBLEM HERE AS SHE WON'T DROP IT EITHER. GRANDMA IS INNOCENT BUT HAS A DAUGHTER AND DAUGHTER-IN-LAW WHO WON'T GROW UP AND DROP IT

**GENERATED SUMMARY:** GRANDMA AND AUNT HAVEN'T SPOKEN IN YEARS OVER A PHONE THAT DIDN'T GET HUNG UP. GRANDMA WANTS TO RECONCILE AND CLEAR THE AIR, BUT AUNT WON'T GO NEAR HER, WON'T LET HER HUSBAND AND KIDS GO THERE, AND AVOIDS.

**REWARD MODEL OUTPUT:** 5.40

---

