# OpenReview forum: "Symmetric Reinforcement Learning Loss for Robust Learning on Diverse Tasks and Model Scales"
_ICML.cc/2025/Conference — ICML 2025 poster_

### Official Review · Reviewer_ANjE · 2025-02-18

**Overall Recommendation:** 3

**Summary:**

This paper proposes a “Symmetric Reinforcement Learning Loss” (SRL) to improve the robustness of policy-gradient algorithms—namely A2C and PPO—when facing noisy or inconsistent advantage estimates.
The core idea is to adapt the concept of “symmetric cross-entropy,” to the RL setting, originally developed for classification under label noise. Concretely, the authors add a “reverse RL loss” term that, together with the standard policy-gradient loss, forms a “symmetric RL loss.” They name the resulting methods SA2C (Symmetric A2C) and SPPO (Symmetric PPO).
The paper demonstrates that in both standard control tasks (Atari, MuJoCo, Box2D) and RLHF-style language tasks (IMDB positive-sentiment generation and TL;DR summarization), the symmetric loss improves performance over standard A2C/PPO—especially when reward signals (or advantage estimates) are noisy or suffer from misalignment.
The authors attribute the gain primarily to PPO’s increased susceptibility to advantage “sign flips” (confusion) and show that introducing the reverse RL term helps correct or dampen these detrimental effects.

**Claims And Evidence:**

- Main Claim: Adding a symmetric term to the actor loss, which mirrors the standard policy-gradient update in reverse, yields more stable and robust learning, leading to higher final performance across diverse tasks and scales. Instabilities are hypothesized to be mainly caused by advantage "sign flips" which are emphasized through advantage normalization based on mini batch statistics.

- Evidence: The paper reports results on 22 Atari games (with and without artificially flipped rewards), several MuJoCo/Box2D continuous-control tasks (with injected noise), and two RLHF tasks (IMDB sentiment and TL;DR summarization). In nearly all tested scenarios with noise or sign confusion, SPPO outperforms PPO, often by a substantial margin. Plots show that advantage sign flips are common, which corroborates the main motivation. In RLHF experiments, SPPO achieves higher model-based reward scores and higher human-like preference judgments (via GPT-4) than PPO.

A different potential fix to the problem of the high fraction of advantage sign flips would be to use significantly larger mini batch sizes. To the best of my knowledge, PPO practitioners have found much larger batch sizes of $\gt 100k$ samples to work much better than smaller batch sizes (Rudin et al. 2022, Learning to Walk in Minutes Using Massively Parallel Deep Reinforcement Learning).

While the results are overall convincing, the paper primarily focuses on final performance metrics (rewards, policy returns, or model-based reward scores) and includes standard deviation ranges across multiple seeds. The paper would benefit from presenting the results visually (training curves or bar charts) rather than in tabular form, which is very hard to interpret well. Further, it could benefit from using more thorough metrics like IQM and confidence intervals aggregated over multiple environments/experiments (Agarwal et al. 2022, Deep Reinforcement Learning at the Edge of the Statistical Precipice).

**Essential References Not Discussed:**

-

**Experimental Designs Or Analyses:**

- Soundness/Validity: The designs follow standard practice in RL research: multiple seeds, standard baselines (A2C/PPO vs. SA2C/SPPO), well-known benchmark environments. The chosen “noise injection” strategies are straightforward and sensible to highlight the method’s robustness. The RLHF experiments use standard reward-model training and GPT-4 preference judgments, which is a recognized approach in alignment research.
- Potential Issues: RLHF tasks rely on pretrained reward models, which can itself be noisy. While that’s precisely the paper’s motivation, it slightly complicates attributions of success (i.e., is the improvement from simply ignoring reward-model mistakes or from genuinely mitigating advantage confusion?). The authors do partially demonstrate examples (e.g., empty summaries yielding higher reward) to show how confusion arises. Overall, the experiments appear valid, though it would be beneficial to see more fine-grained ablations (for example, comparing the effect of bigger vs. smaller batch sizes on advantage confusion).

**Methods And Evaluation Criteria:**

-

**Other Comments Or Suggestions:**

-

**Other Strengths And Weaknesses:**

Strengths:
- Simple, elegant idea that is easy to integrate with existing policy-gradient code.
- Demonstrates robust gains across various tasks (discrete/continuous control, language tasks) and with artificially perturbed rewards and real RLHF-based noise.
- Thorough experiments with multiple seeds and hyperparameter sweeps.

Weaknesses:
- The method adds extra hyperparameters (α, β, Z), which can be a drawback for ease of adoption—though the authors fix α and Z in their experiments.
- While the paper shows that advantage sign flips frequently occur, the discussion is largely heuristic rather than formal.
- For the RLHF tasks, all evaluation rests on an imperfect reward model and GPT-4-based comparison (no direct human eval). Though typical in modern LLM research, more thorough human-based or multi-metric evaluations could be enlightening.

**Questions For Authors:**

-

**Relation To Broader Scientific Literature:**

The work draws work with robust supervised learning under label noise (symmetric cross-entropy). It also builds on well-known RL fundamentals (A2C, PPO) and RLHF techniques. The paper cites relevant prior methods for dealing with overestimation, distribution shifts, and training instability (Double DQN, GAE, trust-region methods). This connection to “robust supervised loss” methods is the main novelty. The authors also mention alternative strategies for RLHF like ranking-based or preference-based RL. Overall, the references and positioning in the literature appear reasonable.

Im the introduction the authors further mention the use of supervised learning techniques in RL (ensembles, layernorm, batch norm), without any citations. These works should be cited:
- REDQ: Chen et al. 2021
- DroQ: Hiraoka et al. 2021
- CrossQ: Bhatt et al. 2024

**Theoretical Claims:**

I did not check the derivations.

---

> ### Author Rebuttal · Authors · 2025-04-01
>
> Thank you for your thorough review and constructive feedback on our paper. To address your questions and concerns, we have provided detailed responses below.
>
> **Question 1**
> > PPO practitioners have found much larger batch sizes of 100k samples to work much better than smaller batch sizes
>
> => Thank you for pointing out this work that uses very large batch sizes. However, the effectiveness of batch size remains somewhat controversial—likely depending on the environment—as other studies [1, 2] argue that smaller batch sizes can be more beneficial. The success of the work you mentioned may be due to its specific setting, where over 10,000 agents can generate a massive number of data points. In general, we believe that, given typical computational and cost constraints, updating an LLM with a batch size of 100k is often not feasible in standard setups.
>
> **Question 2**
> >  Is the improvement from simply ignoring reward-model mistakes or from genuinely mitigating advantage confusion?
>
> => Rather than focusing on reducing the error rate of a trained reward model (which is certainly one way to improve performance), our method directly targets the mitigation of advantage confusion. Even when there is no reward model error, advantage confusion can still arise due to techniques like advantage normalization (Figure 2). We observe that SPPO still shows improvements in the noise-free setting (Section 5.4).
>
> **Question 3**
> > Comparing the effect of bigger vs. smaller batch sizes on advantage confusion
>
> => Thank you for suggesting this interesting experiment. If the paper is accepted, we will include results evaluating our method across a range of batch sizes. However, we believe the effect is quite task-dependent. Based on our current understanding, Atari games tend to perform better with smaller batch sizes [1, 2], while MuJoCo tasks may benefit from larger batch sizes [3].
>
> **Weakness 1**
> > The method adds extra hyperparameters $(\alpha, \beta, Z)$, which can be a drawback for ease of adoption
>
> => While the symmetric RL loss introduces three hyperparameters, $\beta$ and $Z$ can be treated as a single term. To demonstrate ease of adoption, we conducted a hyperparameter sensitivity analysis for $\beta$ while keeping $\alpha$ and $Z$ fixed—particularly in the PPO setting (see Table 11 in the Appendix). We observed consistent performance improvements across a wide range of $\beta$ values, suggesting that the method is easy to apply in practice.
>
> **Weakness 2**
> > While the paper shows that advantage sign flips frequently occur, the discussion is largely heuristic rather than formal.
>
> => We provide empirical evidence of sign flip ratios in Table 2, evaluated across multiple random seeds for a diverse set of environments (5 for Atari games and 30 for the MuJoCo benchmark). We believe that advantage sign changes are obviously  expected after advantage normalization, but we will include results from additional environments to further support this observation.
>
> **Weakness 3**
> > For the RLHF tasks, all evaluation rests on an imperfect reward model and GPT-4-based comparison (no direct human eval). Though typical in modern LLM research, more thorough human-based or multi-metric evaluations could be enlightening.
>
> => We used GPT-4 as the evaluator and measured win rates based on two comparison examples, following the AlpacaEval [4]. Our experiments involve relatively smaller models and simpler tasks, where GPT-4’s evaluations have been shown to align closely with human judgments.
>
> **Thank you for suggesting REDQ, DroQ and CrossQ. We will make sure to cite them in the our paper.**
>
> [1] The Phenomenon of Policy Churn, Neurips 2022\
> [2] Small batch deep reinforcement learning, NeurIPS 2024\
> [3] Sample Efficient Deep Reinforcement Learning via Uncertainty Estimation, ICLR 2022\
> [4] Length-Controlled AlpacaEval: A Simple Way to Debias Automatic Evaluators, COLM 2024

---

### Official Review · Reviewer_7vHN · 2025-02-25

**Overall Recommendation:** 3

**Summary:**

The manuscript introduces a symmetric reinforcement learning (RL) loss function designed to improve the robustness of RL algorithms like A2C and PPO. The proposed symmetric RL loss is inspired by reverse cross-entropy (RCE) used in noisy classification tasks, and the authors apply it to both discrete (Atari) and continuous (MuJoCo, Box2D) RL tasks. The experiments demonstrate that Symmetric A2C (SA2C) and Symmetric PPO (SPPO) outperform traditional A2C and PPO, particularly in environments with added reward noise.

**Claims And Evidence:**

Strengths
1. Strong Experimental Setup: The manuscript provides comprehensive experiments across diverse environments (Atari, MuJoCo, Box2D) and real-world applications (IMDB, TL;DR summarization), demonstrating the generalizability of the approach.
Weaknesses and Areas for Improvement
1. Insufficient Comparison with Other Robustness Methods: While the paper focuses on the symmetric RL loss, it lacks detailed comparisons with other established methods for robust RL, such as Double Q-learning or SAC, which could provide a clearer context for its benefits.
2. Limited Theoretical Justification: The paper would benefit from a more detailed theoretical explanation of why the symmetric RL loss is particularly effective in addressing the noise in reward prediction, especially when compared to existing techniques like Generalized Advantage Estimation (GAE).
3. Hyperparameter Sensitivity: The sensitivity of the performance to hyperparameters such as α and β is not sufficiently explored. More extensive analysis is needed to understand how these hyperparameters affect training and performance stability.
Scalability and Computational Cost: The manuscript does not provide enough information regarding the computational complexity and scalability of the proposed method, especially in large-scale environments or real-time applications.

**Essential References Not Discussed:**

Scalability and Computational Cost
	• Provide a discussion on the computational cost of the symmetric RL loss, especially in large-scale or real-time settings.
	• Analyze the scalability of the approach across different model architectures.

**Experimental Designs Or Analyses:**

4. Experiments
4.1 Atari Games
	• Section 5.1: The results on Atari games show SPPO performing well in noisy settings, but the authors do not discuss the implications of these results for the real-world application of RL algorithms.
	○ Suggestion: Provide insights into how the improvements observed in Atari games can be generalized to real-world tasks, particularly those that involve more complex reward models or environments.
4.2 MuJoCo and Box2D Tasks
	• Section 5.2: While the performance of SPPO is evaluated on continuous action tasks, the impact of reward noise is not thoroughly discussed.
        ○ Suggestion: Include a more detailed analysis of how the added noise specifically affects the training process and how the symmetric RL loss mitigates these effects in continuous action spaces.

**Methods And Evaluation Criteria:**

Clear Results: The experiments clearly show the effectiveness of SPPO over PPO, especially in noisy reward environments.

**Other Comments Or Suggestions:**

1. Abstract
	• Suggestion: The abstract outlines the problem and proposed solution but lacks quantitative details on the performance improvements achieved by SPPO over PPO, especially under noisy conditions.
	○ Recommendation: Include key findings, such as the percentage improvement in reward scores or stability in noisy environments.
2. Introduction
	• Section 1, Paragraph 4: The authors mention that "RL methods introduce challenges such as moving targets and high gradient
          variance," but they do not sufficiently explain how these issues are specifically mitigated by the symmetric RL loss.
       ○ Suggestion: Provide a clearer link between the inherent challenges of RL and how the proposed symmetric RL loss addresses these challenges. Specifically, discuss how reverse cross-entropy aligns with the noise in reward models.

**Other Strengths And Weaknesses:**

Please refer to Claims and Evidence.

**Questions For Authors:**

None

**Relation To Broader Scientific Literature:**

5. Conclusion
	• Section 6, Paragraph 2: The conclusion briefly mentions future work but does not propose specific directions for improving or extending the symmetric RL loss method.
       ○ Suggestion: Discuss potential future improvements, such as exploring the scalability of the method in large-scale environments, or applying the approach to other RL algorithms or tasks (e.g., multi-agent systems).

**Theoretical Claims:**

3. Methodology
3.1 Symmetric RL Loss
	• Section 4, Paragraph 2: The explanation of the reverse RL loss and its application to A2C and PPO is somewhat vague. The connection between the noisy advantage estimates in RL and the reverse cross-entropy loss is not fully established.
	○ Suggestion: Provide a more detailed explanation of how the reverse RL loss functions in practice, particularly focusing on how it corrects the advantage sign confusion caused by noisy rewards.
3.2 Symmetric A2C and PPO
	• Section 4, Equation 9: The use of constants α and β is introduced, but the rationale for their selection is not clear.
        ○ Suggestion: Discuss how the values of α and β are chosen. Are they determined through cross-validation, or are they fixed based on prior empirical knowledge? A sensitivity analysis of these hyperparameters would be valuable.

---

> ### Author Rebuttal · Authors · 2025-03-31
>
> Thank you for your thorough review and valuable feedback on our paper. We hope the response below addresses your concerns.
>
> **Weakness 1**
> > Insufficient Comparison with Other Robustness Methods: While the paper focuses on the symmetric RL loss, it lacks detailed comparisons with other established methods for robust RL, such as Double Q-learning or SAC, which could provide a clearer context for its benefits.
>
> => Rather than expanding to additional types of reinforcement learning algorithms, we focused on those whose working mechanisms align with the symmetric loss formulation in noisy classification settings. Instead, we aimed to demonstrate the effectiveness of the symmetric RL loss across a diverse range of problem domains—including discrete action spaces, continuous action spaces, and NLP tasks—highlighting its applicability even to large language models.
>
> **Weakness 2**
> > Limited Theoretical Justification: The paper would benefit from a more detailed theoretical explanation of why the symmetric RL loss is particularly effective in addressing the noise in reward prediction, especially when compared to existing techniques like Generalized Advantage Estimation (GAE).
>
> => In all our experiments, we used Generalized Advantage Estimation (GAE), and we will explicitly clarify this in the experimental section. Since GAE depends on the reward, noise in the reward can lead to confusion in advantage estimation. Our symmetric RL loss acts as an effective accelerator to mitigate this issue. We provide a detailed analysis of how this acceleration benefits the learning process in Section 4.3 and Appendix A.1 to B.2.
>
> **Weakness 3**
>
> > Hyperparameter Sensitivity: The sensitivity of the performance to hyperparameters such as $\alpha$ and $\beta$ is not sufficiently explored.
>
> => We provide additional hyperparameter sensitivity analysis in Table 11 of the Appendix for SPPO, both with and without reward noise, showing consistent improvements across a range of values. Note that $Z$ and $\beta$ can be treated as a single hyperparameter (see the first paragraph of Section 4.1 for details).
>
> **Weakness 4**
>
> Scalability and Computational Cost: More extensive analysis is needed to understand how these hyperparameters affect training and performance stability.
>
> => The symmetric RL loss is scalable, even for large language models. When selecting the next action (token) via softmax, the model also produces probabilities for non-selected tokens. Computing gradients with respect to these non-selected probabilities introduces some additional overhead, but this cost is roughly equivalent to adding a single MLP layer (we will include this explanation in the paper). Given the overall size and structure of LLMs, this additional cost does not significantly affect training speed. In fact, SPPO showed faster training than PPO for some seeds—i.e., within the overhead of adding one MLP, GPU memory allocation conditions had a greater effect. We briefly address training speed in the last paragraph of Section 5.5.

---

### Official Review · Reviewer_xvu3 · 2025-03-14

**Overall Recommendation:** 3

**Summary:**

Reinforcement learning (RL) training is inherently unstable due to factors such as moving targets and high gradient variance. Reinforcement Learning from Human Feedback (RLHF) and Reinforcement Learning from AI Feedback (RLAIF) introduce additional challenges. For instance, diverse preferences complicate the alignment process, and prediction errors in a trained reward model can become more severe as the LLM generates unseen outputs. These RL challenges create confusion about whether the probability of
an action for a given state should be increased or decreased, similar to the noise in labels for classification tasks. In this work, the authors focus on RL algorithms that share learning difficulties with cross-entropy loss, especially for low-probability predictions. To enhance stability, they adapt reverse cross-entropy (RCE) from supervised learning for noisy data, defining a symmetric RL loss. They demonstrate performance improvements across various tasks and scales.

# Update after rebuttal
All my concerns have been addressed thanks to the authors' efforts in rebuttal. In all, I decided to raise my score to weak accept.

**Claims And Evidence:**

The SPPO applies symmetric loss by adding traditional RL loss with a reverse RA2C or RPPO loss. And the result in Table 1 somewhat validates the claims that SPPO avoids value estimation errors due to the sign of the advantage in advantage normalization dependent on how the batch is composed.

**Essential References Not Discussed:**

N/A

**Experimental Designs Or Analyses:**

Experimental design is valid and analysis is comprehensive.

**Methods And Evaluation Criteria:**

I don't understand why adding the reverse advantage loss could help relieve the problem of estimation error resulting from noisy human feedback and multiple sources of scaled models.

**Other Comments Or Suggestions:**

Add more recent RL baseline comparison in Table 2 and 3.

**Other Strengths And Weaknesses:**

Strength:
- Formulation is clear and the idea combines symmetric cross entropy and A2C/PPO framework.
- It provides theoretical proof of gradient analysis of RL loss and reverse RL loss.

Weakness:
- Motivation is unclear about why bother to use symmetric CE loss other than directly train on large amounts of data sets of diverse tasks.
- Baselines are too few for comparison in Table 2.

**Questions For Authors:**

- What is the role of Z in equation 7.
- Why SPPO still performs well in noisy-free setting.
- What is the benefits of advantage normalization with small batch sizes.

**Relation To Broader Scientific Literature:**

It has a large impacts to the domain of RLHF with noisy reward lables and non-stationary feedback signal.

**Theoretical Claims:**

The authors use gradient analysis to formulate the policy update and show the convergency guarantee in appendix.

---

> ### Author Rebuttal · Authors · 2025-03-31
>
> Thank you for your thorough review and valuable feedback on our paper. We try to handle your questions and provide additional details below.
>
> **Please let us know if our responses resolve your questions and concern. If so, we would greatly appreciate your consideration in updating your score. We’re also happy to continue the discussion if you have further questions. Our answers are as follows:**
>
> **Question 1**
> > I don't understand why adding the reverse advantage loss could help relieve the problem of estimation error resulting from noisy human feedback and multiple sources of scaled models.
>
> => In the Gradient Analysis section, we describe how the reverse RL loss helps the learning procedure in the presence of noise, acting as an accelerator. We believe you would agree that when noise is present, the policy lacks certainty about which action to take. Since the reverse RL loss gradient has its maximum magnitude at 50\% (a parabolic function) and its direction aligns with the original RL loss, it helps the policy move away from this ambiguous state. Please refer to further details in Section 4.2 and Sections A.1 to A.4 in the Appendix. We provide a comprehensive derivation and analysis for all cases.
>
> **Question2**
> > Why bother to use symmetric CE loss other than directly train on large amounts of data sets of diverse tasks?
>
> => Increasing the data amount is one potential solution, but Reinforcement Learning (RL), especially with large models, has burdensome such as generating actions and  getting the corresponding rewards. For the rewards, we would need a highly engineered reward function, human, or AI evaluators (which can also have noise). Therefore, achieving better performance using the same amount of data is obviously advantageous. The symmetric RL loss (or symmetric CE loss) facilitates this and is easy to implement. Since action (or token) sampling is done through the softmax, the other probabilities can be obtained naturally for symmetric losses.
>
> **Question 3**
> > What is the role of $Z$ in equation 7?
>
> => $Z$ was introduced in previous work [1] on noisy classification datasets to address the issue of $\log 0 = -\infty$, which cannot be used when updating a neural network. To resolve this numerical problem, $-\infty$ is replaced with a negative value $Z$, which has been shown to still satisfy the conditions of a robust loss function. In our case, we also use $Z$ to handle negative advantages (see Section 4.1 for more details)
>
> **Question 4**
> > Why SPPO still performs well in noisy-free setting?
>
> => Compared to A2C, PPO has off-policy update parts and typically uses smaller batch sizes (e.g., 64), whereas A2C processes all data points at once (i.e., no batch). Additionally, PPO applies the advantage normalization by default, which can introduce sign changes in the advantages. While PPO is more sample-efficient than A2C, these characteristics introduce sources of noise—even in noise-free settings. Our symmetric RL loss helps mitigate these noisy factors, which can lead to confusion in advantage estimation, thereby improving performance even in clean environments (see Section 5.4 for details).
>
> **Question 5**
> > What is the benefits of advantage normalization with small batch sizes?
>
> => While some studies [2, 3] suggest that smaller batch sizes can be more beneficial than larger ones, our main point is not directly about batch size. In practice, we commonly use batch sizes such as 32, 64, or 512 for PPO, whereas A2C processes the entire dataset without batching.However, after applying advantage normalization, the signs of the advantages can flip depending on how the batch is sampled. For example, a given sample $(s,a)$ might end up with a positive or negative advantage solely based on the composition of the batch. This introduces confusion, which can be interpreted as a form of noise. In such cases, the symmetric loss can help mitigate the impact of that noise.
>
> That said, this doesn’t mean advantage normalization is harmful. Although it can cause sign changes in advantages, it also helps limit the influence of extremely large advantage values, thereby stabilizing the policy update process (Please refer to Section 5.4).
>
> In summary, the symmetric RL loss retains the benefits of advantage normalization while further alleviating the instability caused by sign changes.
>
> **Weakness 1**
> > Baselines are too few for comparison in Table 2.
>
> => We further provide results for PPO in a noise-free setting in Table 12, including an additional environment. Additionally, we report A2C results in Tables 8 and 9 to support further comparison.
>
> [1] Symmetric Cross Entropy for Robust Learning with Noisy Labels, ICCV 2019\
> [2] The Phenomenon of Policy Churn, Neurips 2022\
> [3] Small batch deep reinforcement learning, NeurIPS 2024

---

> > ### Comment · Reviewer_xvu3 · 2025-04-08
> >
> > Thanks for the author's detailed explanation and provide additional comparison results. All my concerns have been resolved and I decided to raise my score accordingly.

---

### Official Review · Reviewer_epYB · 2025-03-17

**Overall Recommendation:** 3

**Summary:**

The paper proposes a new family of loss, symmetric A2C and symmetric PPO loss for RL tasks.

**Claims And Evidence:**

Claims: The paper asserts that its policy gradient formulation with the newly proposed loss achieves superior or at least competitive performance in Atari benchmarks, and that using GPT-J with RLHF on TLDR/IMDB datasets demonstrates improved alignment.

Evidence: For Atari, the evidence primarily comes from charts showing improved scores vs. baseline agents, which seem reasonably convincing if one accepts the standardness of Atari benchmarks. For RLHF, the evidence includes limited experimental data on GPT-J with relatively small and somewhat outdated datasets, making it less clear if the approach scales to more advanced language models and richer feedback datasets.

**Essential References Not Discussed:**

N/A

**Experimental Designs Or Analyses:**

The experiments on Atari appear consistent, using standard protocols (training steps, seeds, reported scores). For the RLHF experiments, the design is somewhat limited: the use of GPT-J with TLDR/IMDB is not clearly motivated given the more up-to-date large language models and more comprehensive feedback datasets available.

**Methods And Evaluation Criteria:**

Methods: The main methodological innovation is the introduction of new loss functions (Equations 8 and 9) that factor in reward signals and the policy gradient. However, the approach enumerates all possible actions in the loss, which can be computationally feasible for small/medium discrete action spaces (e.g., Atari), but becomes very large and inefficient for language modeling.

Evaluation Criteria: Benchmarking on Atari is fairly standard; the paper also attempts to assess alignment improvements on TLDR/IMDB datasets. However, these datasets and the GPT-J baseline may not fully reflect current state-of-the-art for language model RLHF, limiting the generalizability of the reported results.

More recent studies on LLM alignment typically use newer models (e.g., Llama 2 / 3, Qwen 2.5 or beyond) and richer feedback datasets (e.g., Ultrafeedback, Chatbot Arena open data, Nectar etc.). Incorporating these newer data sources and more modern baselines would better situate the method in the current literature and could potentially unlock more significant empirical gains.

**Other Comments Or Suggestions:**

Please see above

**Other Strengths And Weaknesses:**

Please see above

**Questions For Authors:**

Please see above

**Relation To Broader Scientific Literature:**

The paper builds on standard RL methods and extends them with an additional symmetric term.

**Theoretical Claims:**

The derivation looks good to me

---

> ### Author Rebuttal · Authors · 2025-03-31
>
> Thank you for your thorough review and valuable feedback on our paper. We hope the response below addresses your concerns.
>
> **Weakness 1**
> > The approach enumerates all possible actions in the loss, which can be computationally feasible for small/medium discrete action spaces (e.g., Atari), but becomes very large and inefficient for language modeling.
>
> => In large language models, the next token (action) is sampled via softmax, which already provides probabilities for all other (non-chosen) tokens—so there's no need for additional computation to obtain them. Incorporating the probabilities of other tokens during backpropagation introduces some additional computation, but it is roughly equivalent to adding a single MLP layer. Given the overall size and structure of LLMs, this does not significantly impact training speed. In fact, SPPO (the symmetric RL for PPO) showed faster training than PPO for some seeds—i.e., within the overhead of adding one MLP, GPU memory allocation conditions had a greater effect. We briefly address training speed in the last paragraph of Section 5.5.
>
> **Weakness 2**
> > More recent studies on LLM alignment typically use newer models (e.g., Llama 2 / 3, Qwen 2.5 or beyond) and richer feedback datasets (e.g., Ultrafeedback, Chatbot Arena open data, Nectar etc.).
>
> => These methods were motivated from the perspective of noisy rewards in standard deep RL tasks. We show that they perform well across a range of RL tasks and limited experiments in RLHF. It is difficult to cover every setting---we think that the promising nature of this approach is established by our experiments. We promise that if the paper is accepted we will add Qwen2.5 on the TLDR task to the results table.

---

### Decision · Program_Chairs · 2025-05-01

**Decision:**

Accept (poster)

**Comment:**

This paper proposes a novel symmetric RL loss for mitigating noisy reward and preference data problems for RL or RLHF, motivated by the previous reverse cross-entropy loss for classification. The gradient analysis supports an effective acceleration away from abstract states by the reverse RL loss in noisy cases, and experimental results on diverse RL tasks and some small NLP tasks show that the proposed Symmetric PPO (SPPO) improves performances especially for noisy data. The proposed loss seems to be technically sound. The authors sufficiently address most concerns raised by the reviewers, except the lack of empirical validation on RLHF tasks with recent well-known benchmarks and LLMs. Based on the consensus among the reviewers, I would recommend this paper to be accepted. However, the authors need to perform experiments on recent RLHF tasks with cutting-edge LLMs.